# From Next-Token to Mathematics: The Learning Dynamics of Mathematical Reasoning in Language Models

**Shubhra Mishra**[1]     **Gabriel Poesia**[1]     **Noah D. Goodman**[1,2]
Department of Computer Science[1] and Psychology[2]
Stanford University
{shubhra, poesia, ngoodman}@stanford.edu

## Abstract

Large Language Models (LLMs) solely trained on next-token prediction learn to solve a wide range of problems involving mathematical reasoning. But how does this ability evolve during training? We show the first analysis of how mathematical reasoning abilities of several open-weight LLMs develop during pre-training and post-training. To this end, we construct MathCAMPS [1], a synthetic dataset of novel mathematical reasoning problems grounded in 44 fine-grained skills taken from the Common Core curriculum from K to 8th grades. In one experiment, we show that mathematical skills are learned during pre-training in an order that measurably correlates with the human-designed curriculum, even though training data are randomly ordered. We also show a detailed analysis of which mathematical abilities benefit from instruction tuning, a widely used post-training method and, in contrast, which skills suffer. Our work paves the way for an empirical understanding of LLM training dynamics in relation to reasoning.

## 1   Introduction

Large language models are pre-trained on a simple objective of next-token prediction. From this pretext task alone they acquire the ability to perform tasks requiring a surprising range of skills for humans, including answering complex questions, translating between languages, and generating code. Among these tasks, one notable case that has been widely used as a proxy for the general capability of new generations of LLM has been their ability to perform *mathematical reasoning*. This attention to mathematics is natural, since mathematics is a key tool for reasoning underlying many of the most impressive feats of human engineering, such as sending rockets to space or building nuclear reactors. Many benchmarks, such as GSM8K (Cobbe et al., 2021), MATH (Hendrycks et al., 2021), or more recently, FrontierMath (Glazer et al., 2024), have been proposed to evaluate LLMs on mathematical problems of various levels. But how this ability develops *during training* is still poorly understood.

How can we gain insight into the learning dynamics of mathematical reasoning skills? Several open-weights LLMs have been made available with intermediate checkpoints, including Pythia (Biderman et al., 2023), OLMo (OLMo et al., 2025) and Amber (Liu et al., 2023). However, several challenges arise in designing informative evaluations of these models during their training. First, we would like to rule out the potential effect of *data contamination*: LLMs being able to answer a question simply because it was seen during pre-training. Unfortunately, since several of the well-known mathematical reasoning benchmarks have been available on the Web since before these models were trained, this possibility cannot be ruled out by using existing benchmarks such as GSM8K or MATH. Moreover, while these benchmarks are useful for obtaining aggregate estimates of reasoning capabilities, their diversity also precludes a more fine-grained understanding of how specific mathematical skills might evolve or interact. For instance, while "grade-school math" (e.g., in GSM8K) consists of an extremely broad set of specific skills, existing datasets aggregate all of these

---

[1]All code and data associated with the project can be accessed at https://github.com/gpoesia/mathcamps

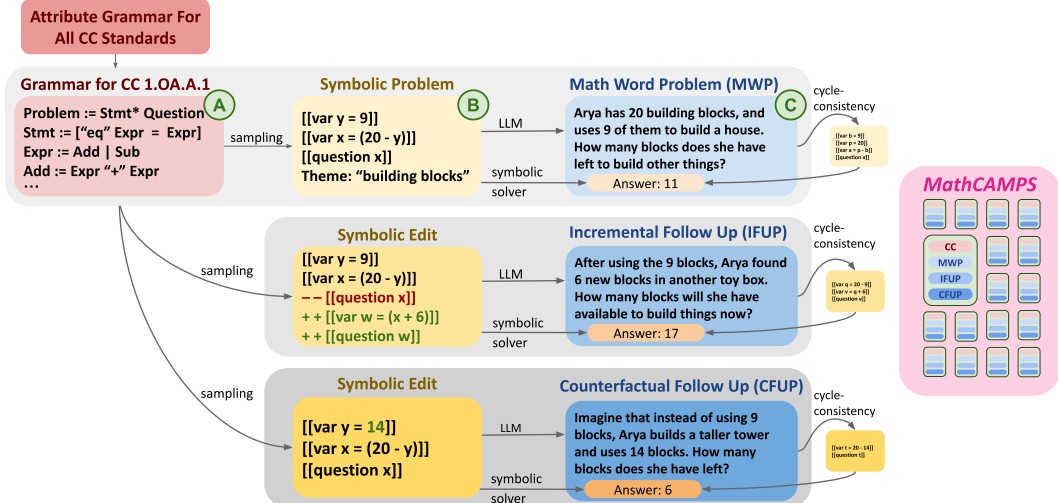

Figure 1: Overview of the MathCAMPS generation pipeline. We start from a grammar (**A**) that represents problems tied to a Common Core Standard - a specific mathematical ability drawn from a human curriculum. We sample problems in a symbolic form (**B**), and use a language model to realize it in natural language (**C**), applying a cycle-consistency where we back-translate the problem into symbolic form and ensure the answer remains the same, validating truthfulness. We also synthesize incremental and counterfactual follow-up problems

into a single final accuracy number. This is useful for comparing different models, but less informative about what exactly models learn or how they might differ.

In this work, we develop a synthetic dataset and design evaluations to address these questions, and elicit a number of insights into how mathematical skills develop during LLM pre- and post-training. Concretely, we introduce MathCAMPS: a fine-grained dataset created automatically from the Common Core (CC) mathematics curriculum, including 49 different skills (CC "standards") from grades K-8. The CC curriculum is adopted by thousands of K-12 schools and includes grade-wise standards that indicate specific skills students must be proficient at by grade. By constructing MathCAMPS in direct relation to the CC, our benchmark enables rich set of analyses of mathematical proficiency in language models, allowing direct parallels to abilities that human students are also evaluated on.

We encode the family of problems associated with each CC standard as a grammar, allowing us to sample arbitrarily many symbolic problems, which our pipeline then converts into natural language word problems using GPT-4o (OpenAI et al., 2024). To ensure that the word problems are faithful to the symbolically generated structures, we introduce a method for cycle consistency that allows us to ensure data quality in a fully automatic manner.

Our fine-grained dataset then allows us to study the training dynamics of several *open-weight language models* for which training checkpoints have been made available. We report on a series of novel analyses of the learning dynamics of mathematical reasoning skills that our methodology enables. Our contributions are:

1. We construct MathCAMPS, a dataset of 4900 new problems stratified into fine-grained capabilities defined by the Mathematics Common Core Standards for K-8 grades. We also release our pipeline to generate arbitrarily many more.

2. We use OLMo2, Amber, and Pythia (families of open-weight models) to analyze how specific mathematical skills evolve during pre-training. We explore alignment between LLM and human learning, skill evolution patterns, and the robustness of reasoning abilities.

3. We use OLMo2 and Amber models to explore how instruction tuning, a widely used post-training method, affects specific mathematical abilities. We find that the specific instruction tuning methodology used in each model varies widely into how broadly beneficial they are to reasoning, as well as which specific mathematical skills they affect the most — both positively and negatively.

## 2 Related Work

Our work closely relates to (i) current benchmarks of mathematical reasoning in LLMs, (ii) benchmarks constructed using LLMs, (iii) behavioral testing and applications in NLP, and finally (iv) open-weight LLMs that provide training checkpoints.

**Benchmarks of mathematical reasoning** MATH (Hendrycks et al., 2021) and GSM8K (Cobbe et al., 2021) have been two leading benchmarks for the evaluation of mathematical reasoning in LLMs. Both datasets consist entirely of human-authored problems — a process that is expensive to reproduce — and as a result, neither benchmarks were updated since their initial releases. Given that LLMs are trained on Web data, it is unclear whether they might have been trained on the test problems of these benchmarks (Bubeck et al., 2023) – either directly or from other sources (e.g., all problems in MATH come from past public competitions). In fact, GSM1K (Zhang et al., 2024), a new dataset that independently attempted to reproduce the data distribution of GSM8K, has found reduced performance on several models, suggesting the possibility of test set contamination. Moreover, these datasets evaluate a very wide range of abilities, without distinction. The GHOSTS dataset provided the first fine-grained evaluation of undergraduate-level mathematical skills in LLMs (Frieder et al., 2023); but given that the reasoning evaluation was performed manually by expert annotators, this unfortunately does not scale to evaluating a large number of training checkpoints. In contrast, the evaluation in MathCAMPS is both fine-grained and fully automated, albeit at the grade school level.

**LLM-generated synthetic datasets for LLMs** As collecting data from human annotators at scale is expensive (especially in domains requiring expertise), prior work has relied on LLMs to aid the generation of large-scale benchmarks (Hartvigsen et al., 2022). BigToM (Gandhi et al., 2023), a benchmark of social reasoning in LLMs, applied the idea of symbolically scaffolding questions for the LLM to realize in natural language, an approach that we transport to mathematics. Dyval (Zhu et al., 2024) proposed a method for generating reasoning problems for LLMs based on a DAG representing the computation. While Dyval contains two mathematical tasks (arithmetic and solving linear equations), MathCAMPS takes this idea further for mathematical reasoning, spanning 44 skills directly grounded on a human curriculum. Other synthetic evaluations focused on mathematical skills include GSMore (Hong et al., 2024) and the concurrent work on GSM-Symbolic (Mirzadeh et al., 2024). Both these works focus on evaluating the robustness of LLMs by *perturbing* existing problems from an existing dataset, GSM8k, whereas in MathCAMPS we synthesize problems from scratch, grounded on a human curriculum (Hong et al. (2024) also proposes perturbations to coding problems, which we do not focus on here).

**Behavioral testing in NLP** Our goal to provide a fine-grained evaluation of mathematical reasoning has parallels with *behavioral testing* — the idea of testing software systems on specific features, as opposed to just their overall adequacy (Ribeiro et al., 2020). In particular, CheckList (Ribeiro et al., 2020) allowed testing machine translation models for fine-grained failure modes. Dynaboard (Ma et al., 2021) proposed an NLP leaderboard where users can adapt to their own needs by choosing the utility of different metrics; our dataset enables a similar user-customizable comparison between models for mathematical reasoning.

**Open-Weight Language Models** Our analysis of the learning dynamics of is done on top of models for which training checkpoints are made publicly available. These include Pythia (Biderman et al., 2023), OLMo (OLMo et al., 2024), and LLM360 (Liu et al., 2023). Moreover, OLMo and LLM360 also have post-trained (instruction-tuned) versions, allowing us to also understand the impact of this step in the LLM training pipeline.

# 3 MathCAMPS

We now describe our pipeline for automatically generating mathematical problems and follow-up questions that are grounded in a human curriculum – the Mathematics Common Core (https://www.thecorestandards.org). Figure 1 overviews our pipeline. We describe the Common Core, how we represent its standards in a grammar, sample symbolic problems, generate follow-ups, realize those in natural language, and finally improve quality by checking for cycle consistency.

## 3.1 The Mathematics Common Core

To ground problems in a human curriculum, we turn to the Common Core State Standards for Mathematics. 41 states in the United States adopt the CC as their curriculum. The CC details the mathematical content that students should master from Kindergarten up to 12th grade. Within each grade, the CC elaborates a series of individual *standards*, which detail a particular mathematical skill that students should learn at that grade. Each standard has an identifier, such as K.CC.C.7, and a summary description — for K.CC.C.7, this is "Compare two numbers between 1 and 10 presented as written numerals". Here, K indicates that this is a standard for the Kindergarten grade level, whereas 8.EE.C.8 — "Analyze and solve pairs of simultaneous linear equations" — is an 8th grade standard.

We take 44 standards spanning grades K through 8 to compose MathCAMPS, focusing on standards that are amenable to automatic problem generation with a final answer in text form. The complete CC curriculum has 229 standards across grades K through 8, bringing our coverage to 19.2% of the curriculum for these grades. Notably, we currently do not cover standards focusing on conceptual understanding (e.g., 3.OA.D.9 – "Identify arithmetic patterns [...], and explain them using properties of operations."), or standards that emphasize visual reasoning (e.g., 6.G.A.4 – "Represent three-dimensional figures using nets made up of rectangles and triangles, and use the nets to find the surface area of these figures."). All 44 standards covered in MathCAMPS are listed in Appendix C.

**Representing Common Core standards**   We represent CC standards as non-terminals in an *attribute grammar* (Heine & Kuteva, 2007) — a rich formalism that can encode semantic, context-sensitive rules. Attribute grammars can encode syntax much like a context-free grammar, but also allow us to embed information processing (e.g., setting and testing conditions on attributes, such as bounds on constants) in the production rules. We map each standard $s$ to a non-terminal $P_s$, such that all strings produced by expanding $P_s$ using production rules are valid symbolic representations of a problem pertaining to standard $i$. Figure 1 shows a (simplified) grammar for the standard 1.OA.A.1 – "Use addition and subtraction within 20 to solve word problems involving situations of adding to, taking from, putting together". Here, a word problem, generated by the Problem non-terminal, consists of a *sequence* of declarative statements expressing equations between expressions. For this standard, an expression consists of addition, subtraction, variables, and constants. After these declarations, the problem ends with a *question* — an expression representing the value that the problem asks for. Concretely, our grammar is implemented in Python: each non-terminal becomes a stochastic function that samples and applies a production rule, recursively expanding non-terminals that it produces. In the grammar in Figure 1 (A), sampling a Problem generates a structure such as the one shown in Figure 1 (B).

**Enforcing problem constraints**   When sampling problems, there is no a priori guarantee that all generated statements are necessary to answer the question. To avoid such statements, we remove them by applying a simple graph reachability algorithm on a dependency graph between statements, removing statements that the answer does not depend on. This enforces the constraint of only having useful statements in problems. Besides this constraint, which we always enforce, each standard can apply specific constraints. The standard 1.OA.A.1 has an example of such constraint: it requires that students only be asked to use "addition and subtraction within 20." To be faithful to this standard, we must validate that no intermediate values used in the solution exceed 20. To encode this and other constraints across the curriculum, we implement a suite of 6 parameterized filters (detailed in Appendix D)

that are selectively applied depending on the standard's specification. Applying rejection sampling from the grammar using the standard's filters gives a procedure for generating valid symbolic *problems*. For all standards that can be formulated as solving a system of linear equations, we use SymPy (Meurer et al., 2017) to obtain final answers. For other cases, we use two simple custom procedures (to list the factors of numbers and to compare values).

### 3.2 From symbolic to word problems

To realize the symbolic problems into natural language, we use few-shot prompting with GPT-4 (Figure 1 (C)). For each standard, we sampled two valid symbolic problems and manually wrote a problem in natural language that faithfully represents the symbolic structure. For standards involving word problems, which typically contain a simple cover story, we also sampled a random theme out of 188 that we crafted (e.g., "Book", "Pirate ship", "Money"). These examples are then given to GPT-4 in-context, along with a new symbolic structure (and a random theme, for standards where that is relevant), requesting it to generate a faithful natural language problem for that structure.

Unlike generating problem stories from a fixed set of templates, using a language model for generating natural language problems gives us fluid, diverse language. Unfortunately, we also lose any guarantee that the generated word problem represents the original symbolic structure faithfully. To mitigate this issue, we also introduce a *cycle consistency* method that we have found to drastically improve problem quality. Precisely, we use the same few-shot examples we crafted for each standard *in reverse* (i.e., with the natural language problem coming first, followed by the symbolic structure) to have GPT-4 translate the word problem it wrote into a symbolic structure. In this step, the model is not given the original structure. We then parse and apply the appropriate solver to the generated symbolic problem; we consider the generation *cycle-consistent* if the answers to the original and recovered problems are the same (illustrated in Figure 1). We then discard problems that fail this test. A more in-depth analysis of the efficacy of cycle-consistency can be found in Appendix F.

### 3.3 Generating follow-up questions

As human instructors know, follow-up questions are often a useful way to probe a student's understanding. In MathCAMPS, we leverage our symbolic representation of problems to derive follow-up questions. We propose two kinds of questions: *counterfactual* questions, where we change a constant in the original problem, and *incremental* questions, where we add a new piece of information. For each CC standard, we mark which (if any) of these two categories of follow-ups are applicable. Symbolically, follow-up questions are represented as a *difference* to be applied to the original question — when we apply the difference, we obtain a new problem. We then use the same solver as the original problem to obtain the ground-truth answer to the follow-up question. We employ the same few-shot structure to translate this difference into a natural language question, and parse it back into a symbolic structure to test for cycle consistency.

## 4 Results

We use MathCAMPS to evaluate how the ability to reason mathematically evolves during pre- and post-training. We use open models that have released intermediate checkpoints (namely Amber from LLM360, OLMo2 7B from OLMo, and Pythia 12B from Eleuther AI). Then, we evaluate a subset of the available checkpoints on MathCAMPS. We use the evolution of checkpoint accuracy on the whole dataset and on subsets of the dataset to measure how different mathematical reasoning abilities evolve during training. Additionally, OLMo2 and Amber have instruction-tuned versions: we use these and Llama-3.1-8B's versions, compared to their based pre-trained checkpoints, to study the impact of instruction tuning on individual Common Core standards. We also perform a comprehensive fine-grained (overall, per-grade, per-standard) evaluation of 23 popular open (including Llama 3, DeepSeek, Mistral of various sizes) and closed (e.g. GPT-4o, Claude 3, Gemini 1.5) models on MathCAMPS for which checkpoints are not available, and report all the results

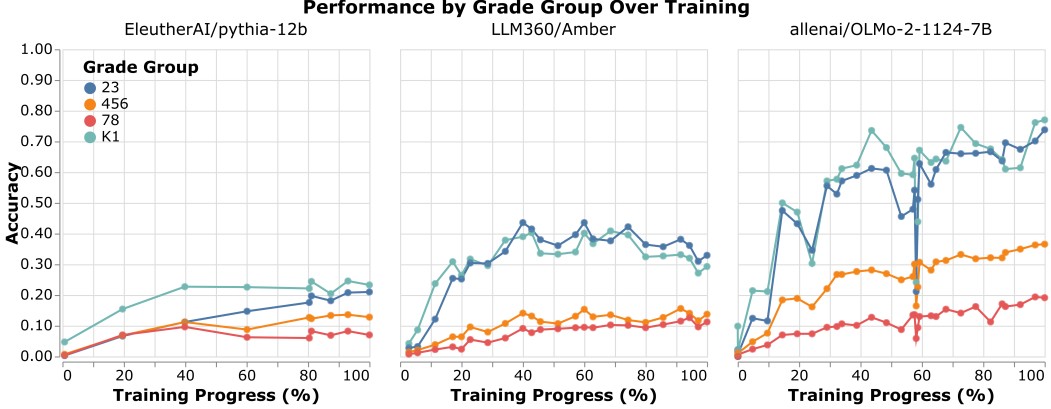

Figure 2: Model accuracy on problems coming from different grade groups evaluated during training. Each data point corresponds to an LLM checkpoint evaluated on MathCAMPS problems testing skills from the indicated range of grades. Training Progress (X-axis) is measured by percentage of total pre-training tokens seen by the checkpoint. Accuracy is final-answer accuracy on solving the problems with few-shot CoT prompting.

in the Appendix H. In the remainder of this section, we present our research questions and observations.

## 4.1 Pre-training

**RQ1: How aligned are the learning trajectories of mathematical reasoning in LLMs compared to human students/curricula?** We first analyze how performance on skills from groups of grades evolve over time. Given overlap in the content for each grade, we split the grades K-8 into K-1, 2-3, 4-6, and 7-8. Then, we analyze how performance on these grade groups improves over time. We see the performance of the OLMo, LLM360, and Pythia models in Figure 2.

Notably, these models' training data is not ordered according to any human curriculum. Despite this, we see that skills from earlier grades show higher performance earlier on in training, and this trend continues. It is possible that LLMs learn skills that are easier for humans first because these skills implicitly build on each other in the human reasoning data that LLMs observe. We see some evidence for this: when we look at the evolution of skills by grade groups, we see that the evolution of these groups follow the human curriculum, since on average, skills from earlier grades are also learned faster. However, even though the aggregate trend holds true, we also find that specific skills are learned in ways unaligned with the human curriculum. For example, 6.EE.A.1, a skill that asks students to *evaluate numerical expressions involving whole-number exponents* is learned very quickly, simply because calculating small exponents for small bases is an easily memorizable task. In contrast, even though human students could also learn these skills by memorization early on, they tend to learn them later only because they're exposed to them later. Another important note is that we only evaluate the ability to solve "computational" problems using these skills, whereas human students also need to pick up more conceptual skills (like explaining exponentiation, making analogies, illustrating numerical scenarios, etc). These abilities are not captured by simply memorizing the results of operations for small numbers.

Next, we analyze the relationship between model performance and the magnitude of the numbers involved in the problem. In Figure 3, we show the accuracy for each training checkpoint for the different numbers of digits seen in the answers. Interestingly, we note how model behavior does not follow exactly what is seen in humans. For example, even though problems with 1-digit answers are easier on average than those with two-digit answers, Amber seems to consistently over perform in cases where there are 2-digit answers.

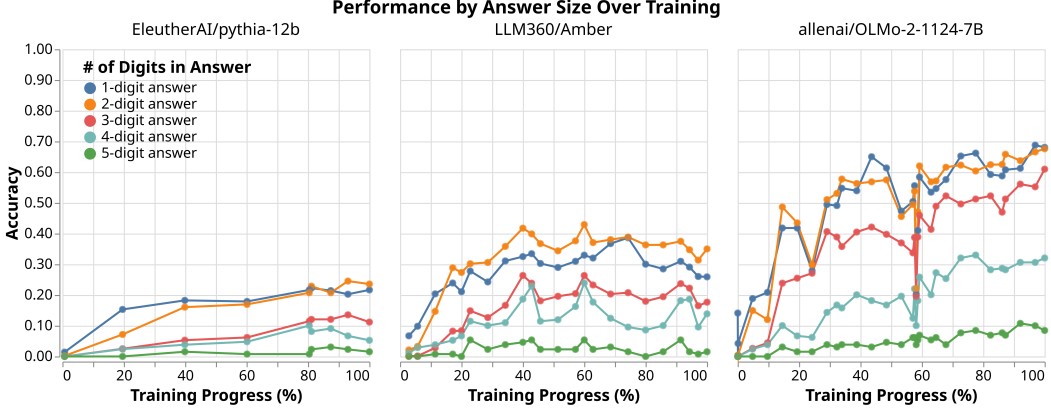

Figure 3: Performance on problems of varying number of digits in their final answer, across pre-training checkpoints.

This holds true for the final Pythia model as well, but not for most other training checkpoints we looked at. This highlights a discrepancy that could potentially reveal underlying mechanisms that prevent LLMs from reasoning robustly and consistently.

**RQ2: Do skills emerge at a certain point in training, or do they evolve smoothly?** We evaluated each checkpoint on a diverse set of 49 skills across grades K-8. We observed that all skills tend to evolve smoothly during training, as opposed to certain skills disctretely emerging at a specific training step. In Figure 4, we show how all skills across grades 2 and 7 evolve across the Pythia, Amber, and OLMo models. Given space constraints, we include results for all other grades in Appendix B.

The granularity in MathCAMPS lets us conduct analyses that aren't possible by looking at aggregate accuracies from datasets like GSM8K/MATH over time. For instance, the skills we included in 2nd grade mostly involve adding and subtracting within 100 under different settings (multi-step word problems, problems including currencies and lengths, etc.). Pythia demonstrates a particular performance gap between standards 2.OA.A.1 and 2.NBT.B.6. The prior encompasses the skill to solve one- and two-step problems with unknowns in all positions and the latter asks that students be able to add up to four two-digit numbers. A higher performance on 2.NBT.B.6 indicates that models performed well in a setting with more additions, but the linguistic complexity of 2.OA.A.1 made the problems harder (despite the numerical calculations in 2.OA.A.1 problems being easier). Amber and OLMo also demonstrate this performance gap in a smaller magnitude.

For seventh grade, however, there is more overlap in which skills models struggle with and which ones they are consistently performant on. 7.NS.A.1 is the skill about adding and subtracting rational numbers, with the fraction and decimal suffixes representing the fractional and decimal subsets of that standard. 7.NS.A.2 is about the multiplication and division of fractions, and 7.NS.A.3 involves solving real-world math problems that include the four operations and rational numbers. Interestingly, we see that across all models, skills involving operations with decimals are learned earlier on and tend to evolve faster. This is likely because operations with decimals are easier to memorize, given their high surface similarity to operations with integers (e.g. 0.28 + 0.45 is very similar to 28 + 45).

However, the skills involving fractions are learned at slower speeds, if they are learned at all. This is likely because operations with fractions require more reasoning steps per operation (e.g. calculating $\frac{7}{3} + \frac{2}{5}$ first requires finding a common denominator before conducting the easier step of addition). An alternate explanation involves the tokenization of numbers: it is likely that decimals and fractions are tokenized differently, and that the tokenization of decimals is more similar to that of integers. This would enable an easier transfer of mathematical reasoning skills learned for integers to skills applied to decimals.

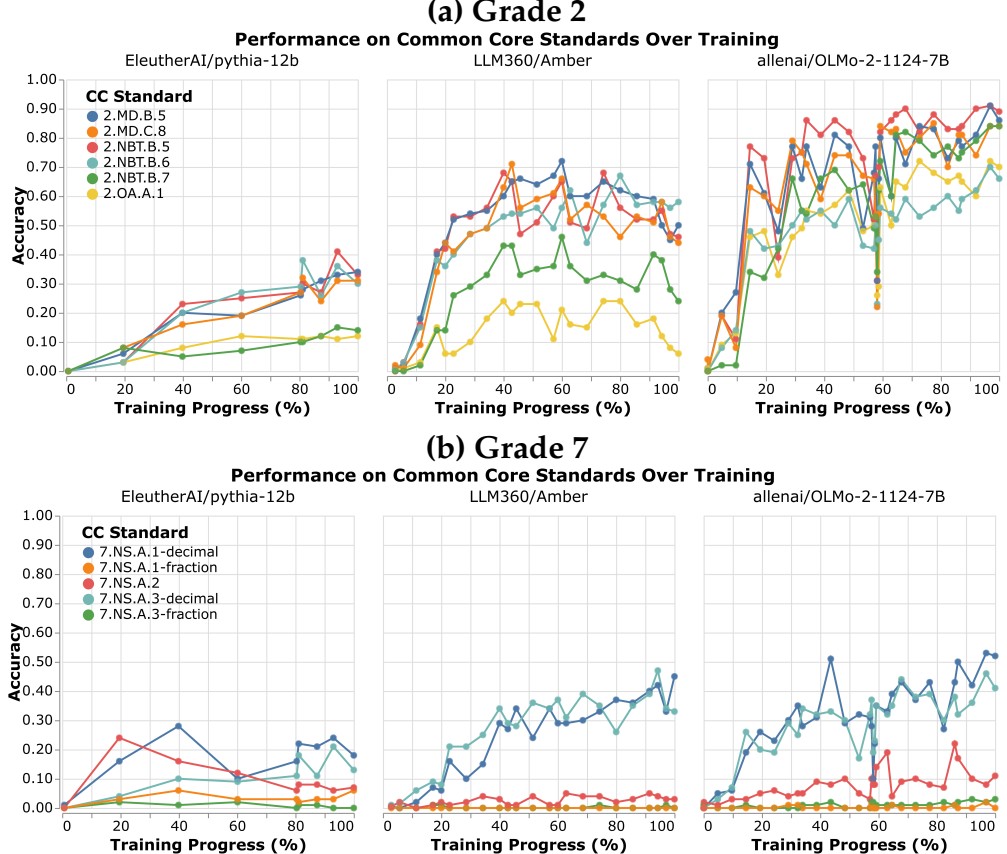

Figure 4: Learning dynamics of individual Common Core standards in grades 2 and 7. Full results for all grades can be found in Appendix B.

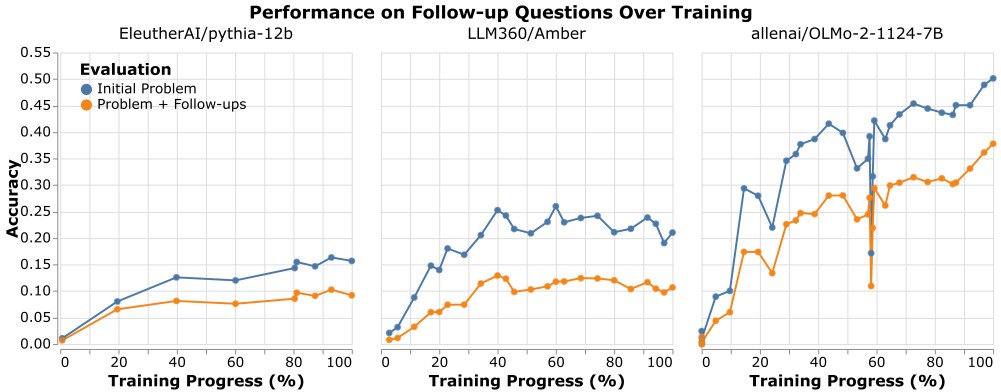

Figure 5: Performance for models during training when also asked to answer follow-up questions about each problem. Here, we only consider problems that have at least one associated follow-up question (counterfactual or incremental).

Additionally, we hypothesize that in some models, skill 7.NS.A.2 is learned relatively quickly because multiplying and dividing fractions requires fewer reasoning steps than adding and subtracting would.

These types of fine-grained analyses enabled by MathCAMPS might inform model pre-training and evaluation, especially for models that are trained with specific end uses in mind (e.g., in educational applications).

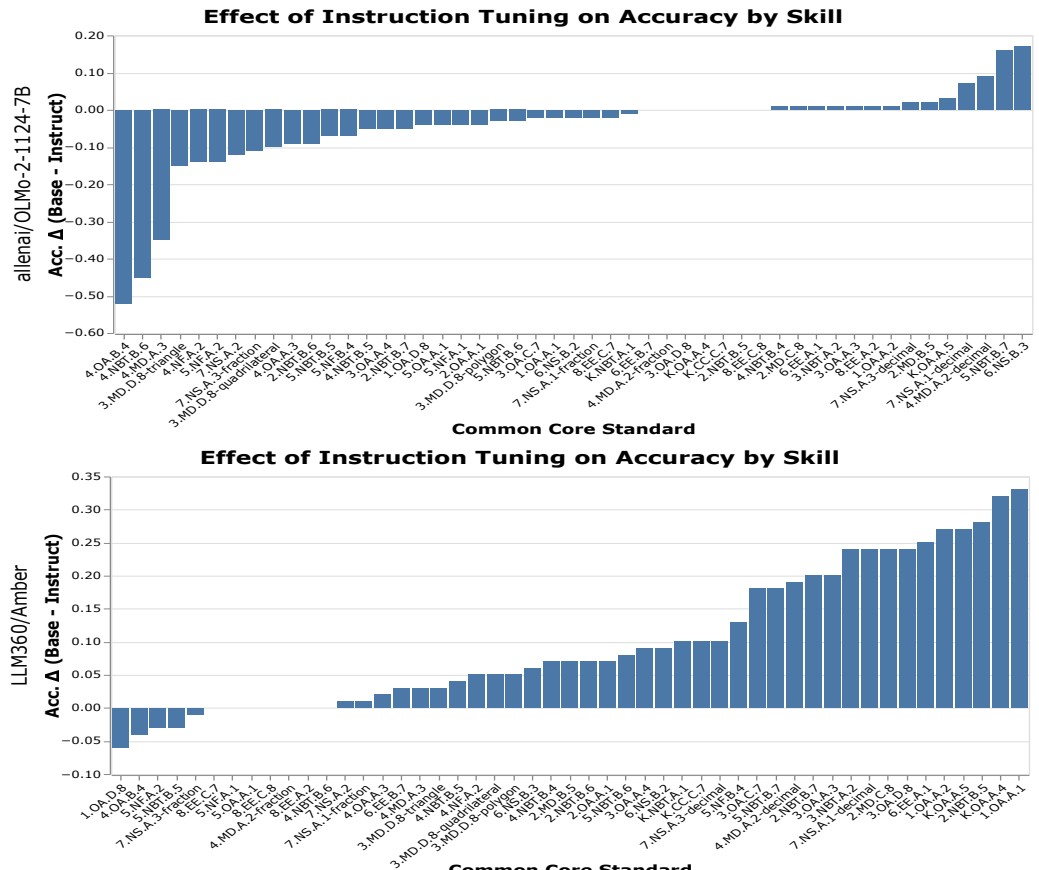

Figure 6: Effect of instruction tuning on individual skills in OLMo and LLM360. We show the difference in accuracy between the base and post-trained model: a negative value means that, on that standard, post-training *improves* performance, whereas a positive value indicates that performance was hurt. We find widely different profiles between the two instruction tuning methods: while the majority of skills in OLMo improve, many more are either unaffected or hurt by the instruction tuning method applied to LLM360.

**RQ3: Do models learn to apply reasoning skills in a robust manner?**  We evaluate whether models' reasoning skills are robust (and not a result of surface-level reasoning) by asking follow-up questions about each problem, probing for deeper understanding (Section 3.3 explains how we generate such questions). Figure 5 shows performance on problems and follow-up questions as training progresses. We observe that generally, as model ability to solve problems improve, so does their ability to answer follow-up questions about the problems they solve, demonstrating a relatively robust capability increase. However, the gap between the ability to answer one-turn and follow-up questions tends to increase over time. It is likely that this isn't caused by a worsening ability to reason deeply, but rather because models solve more and more challenging problems as they train, and problems that are challenging are likely to have follow-ups that are also harder. Overall, we find a surprisingly robust ability to reason about mathematical problems developing during LLM training: not only the problems we use are novel, the models are increasingly capable of answering subsequent questions about those problems — a task we do not generally see in existing mathematical reasoning datasets, nor in post-training datasets focused on conversations (e.g., UltraChat (Ding et al., 2023)).

### 4.2 Post-training

**RQ4: How does instruction tuning impact reasoning abilities?** Here, we use the Amber, OLMo, and Llama models to show how instruction tuning impacts specific mathematical skills. In Figure 6, we highlight Amber and OLMo. We hypothesize that the impact of instruction tuning on a model's overall reasoning ability depends on the base model's reasoning ability. We see that OLMo's reasoning abilities were impacted more positively as compared to Amber. This pattern of predominantly positive impact also holds for Llama-3.1-8B (Figure 7 in the Appendix). We note two potential reasons for this observation. First, the base Amber model is weaker than base OLMo and Llama, which means that it likely doesn't have the latent mathematical reasoning abilities instruction-tuning would let us observe (by enabling the LLM to answer questions). Second, for OLMo specifically, pre-training consists of two stages with distinct data distributions, with the first stage including 3.9T tokens of broad web-scale data and the second stage including 843B tokens of focused, high-quality mathematical and scientific data. Since this second pre-training stage included high quality reasoning data, it improved the base model's ability to reason mathematically further, giving instruction tuning even more of a chance to surface the gains from this second stage of training.

These results contrast with an observation made by the concurrent work of Springer et al. (2025), which measured the impact of the quantity of pre-training data on performance changes due to fine-tuning. They show that the more tokens an LLM sees during pre-training, the less effective instruction tuning is on it. In our case, OLMo and Llama, which were trained on 4T and 15T tokens during pre-training respectively, gained more from instruction tuning compared to Amber, which was only pre-trained on 1.3T tokens.

Although there is a broad variety in how specific skills are impacted by instruction tuning, we note a few interesting trends. First, we see how skills that are extremely challenging for the base model tend to stay unimpacted by instruction tuning. For example, 8.EE.C.8, a skill that measures the ability to solve systems of two equations in two variables, sees no improvement across all three models we inspect. Additionally, we also see some correlation in which skills improved or were hurt by instruction tuning across models. For example, in Figure 8, we see that skills that show improvements after instruction tuning tend to rely on multi-step reasoning, which possibly emerges due to better language understanding and instruction following. Overall, we find that instruction tuning has a variable impact on mathematical reasoning abilities of LLMs, with a nuanced interaction between model, specific skill, and the impact derived from post-training.

## 5 Conclusion

We introduce MathCAMPS, a fine-grained synthetic benchmark of mathematical reasoning in LLMs. MathCAMPS is directly grounded in the Common Core Standards, a widely used curriculum in human education. By tying our problems to a human curriculum, we enable a much wider range of analyses to understand how mathematical reasoning abilities evolve. We show analyses of alignment between human and LLM learning, the evolution of skill learning over training, robustness of reasoning skills over training, and the impact of instruction tuning on specific skills, though we believe these are only a few examples of analyses that MathCAMPS permits. MathCAMPS can also be used to directly compare how humans and LLMs acquire skills over time by measuring the performance of human participants of various ages on MathCAMPS.

While we currently cover 44 CC standards, our pipeline can be easily extended to cover additional standards where problems have a computational nature, and where answers can be obtained using a computer solver. These can include topics beyond high-school, including calculus and linear algebra. This framework, however, is difficult to extend to more conceptual problems, including mathematical proofs, or problems that require *explanations*, as opposed to a final computational answer. Judging natural language reasoning reliably, in the absence of an exact answer to compare to, remains an open problem — an important challenge to allow us to extend the scope of evaluation of mathematical reasoning in LLMs.

## Acknowledgments

This work was supported by a NSF Expeditions Grant, Award Number (FAIN) 1918771. GP was also supported by the Stanford Interdisciplinary Graduate Fellowship (SIGF).

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

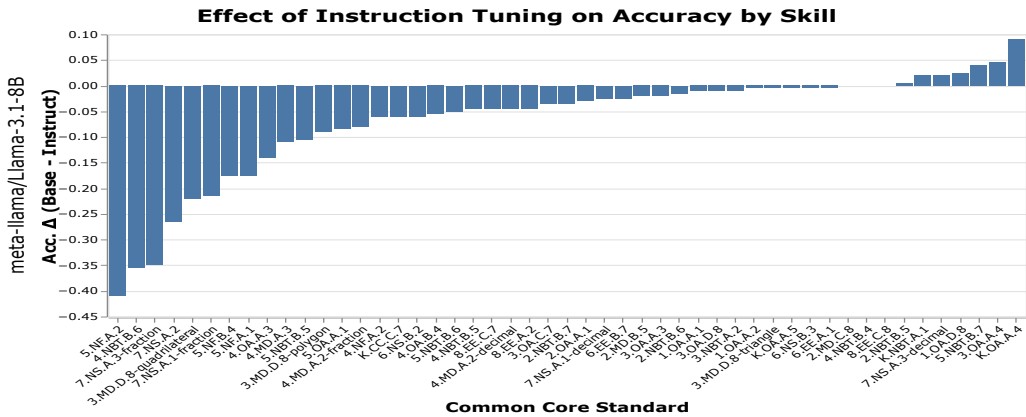

Figure 7: The impact of instruction-tuning on Llama 3.1 8B.

| Standard ID | Description |
|---|---|
| K.CC.C.7 | Compare two numbers between 1 and 10 presented as written numerals. |
| K.OA.A.4 | For any number from 1 to 9, find the number that makes 10 when added to the given number, e.g., by using objects or drawings, and record the answer with a drawing or equation. |
| K.OA.A.5 | Fluently add and subtract within 5. |
| K.NBT.A.1 | Compose and decompose numbers from 11 to 19 Into ten ones and some further ones, e.g., by using objects or drawings, and record each composition or decomposition by a drawing or equation (e.g., 18 = 10 + 8); understand that these numbers are composed of ten ones and one, two, three, four, five, six, seven, eight, or nine ones. |

Table 1: CC Standards for Grade K

## A    Instruction-Tuning Results

In Figure 7, we see the results for instruction tuning on Llama-3.1-8B and in Figure 8, we see a side-by-side comparison of the impact of instruction tuning on all skills.

## B    Learning Dynamics for Grades K-8

Figures 9 through 17 show the learning dynamics for each grade for Amber, LLM360, and OLMo.

## C    Common Core Standards in MathCAMPS

We will be releasing the code used to generate MathCAMPS and our first dataset. All of the Common Core standards we implement will be described in a configuration file, commoncore.yaml, where standards are instantiated by composing high-level components from the Common Core attribute grammar. Moreover, we provide our prompts used to generate the dataset and model responses, as well as all problems and model responses for all LLMs we evaluated.

We list the Common Core standards we represent in MathCAMPS in Tables 1 through 9, segregated by grade. Standards 3.MD.D.8, 4.MD.A.2, 7.NS.A.1, and 7.NS.A.3 are split up into sub-standards. This was done for ease of implementation of the grammar.

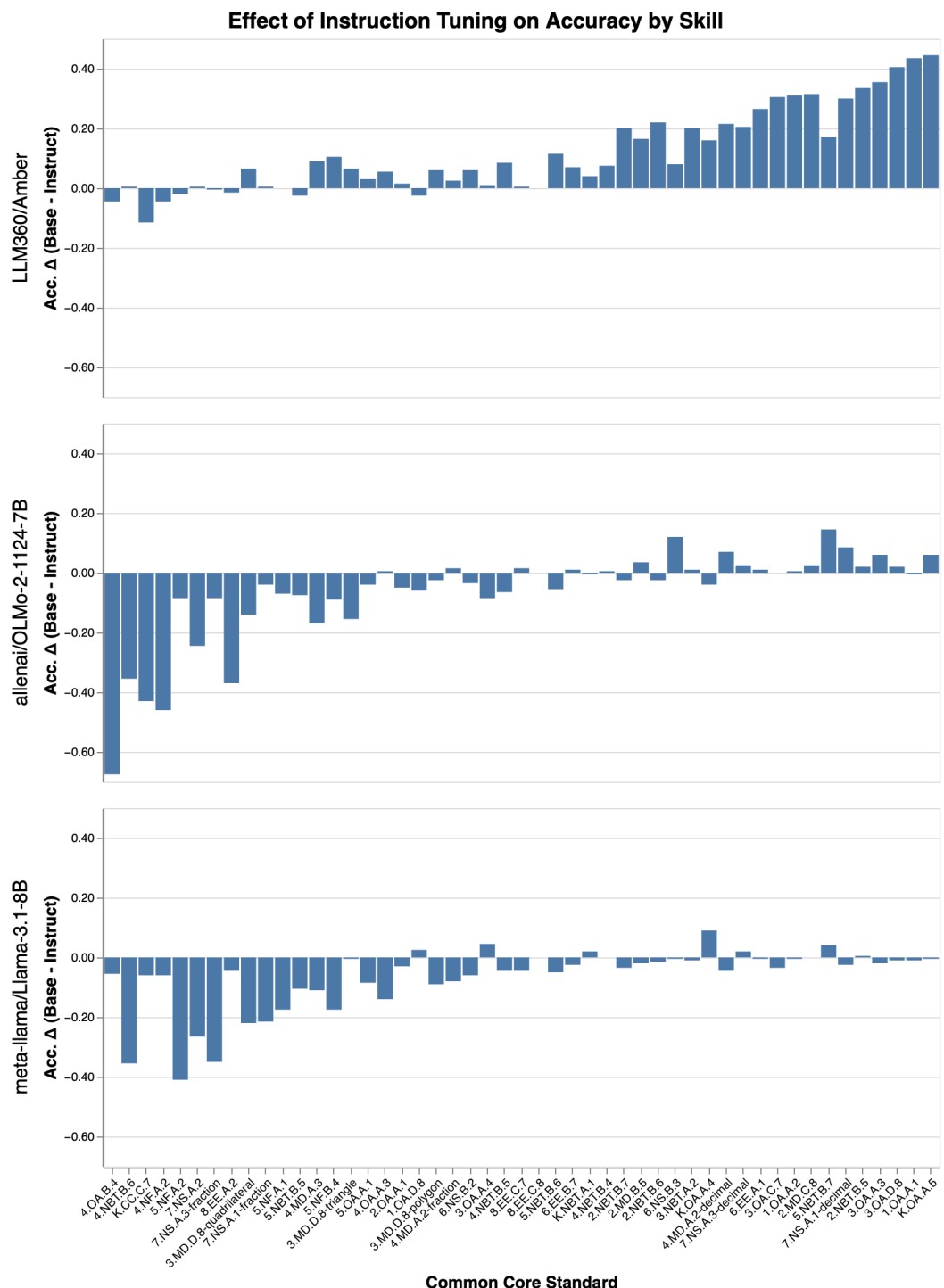

Figure 8: Impact (base minus post-trained performance) of instruction tuning across Amber, Llama, and OLMo, sorted by skill rather than by the performance delta. Overall, the direction of impact by skill tends to be similar across models, even if magnitude differs.

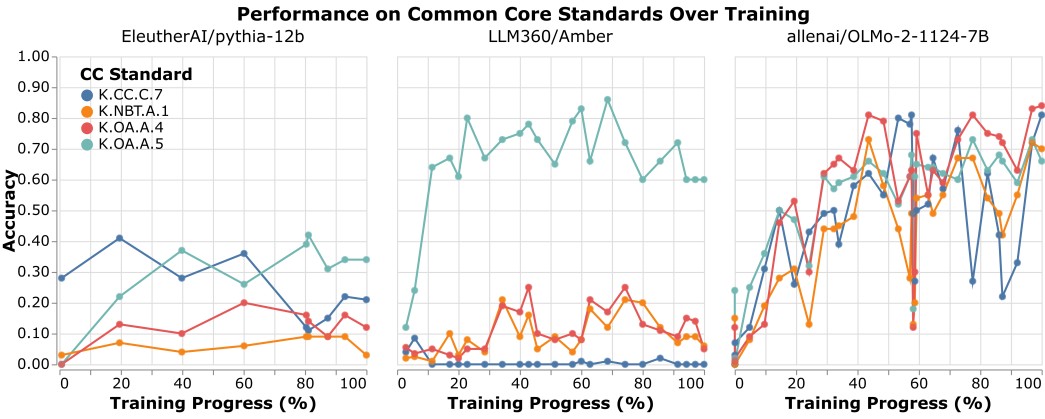

Figure 9: Learning Dynamics Across Amber, Pythia, OLMo for Kindergarten

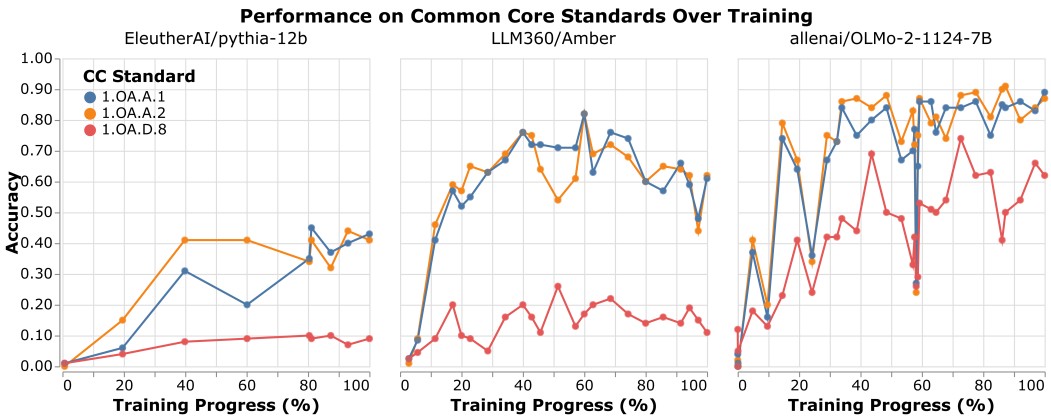

Figure 10: Learning Dynamics Across Amber, Pythia, OLMo for First Grade

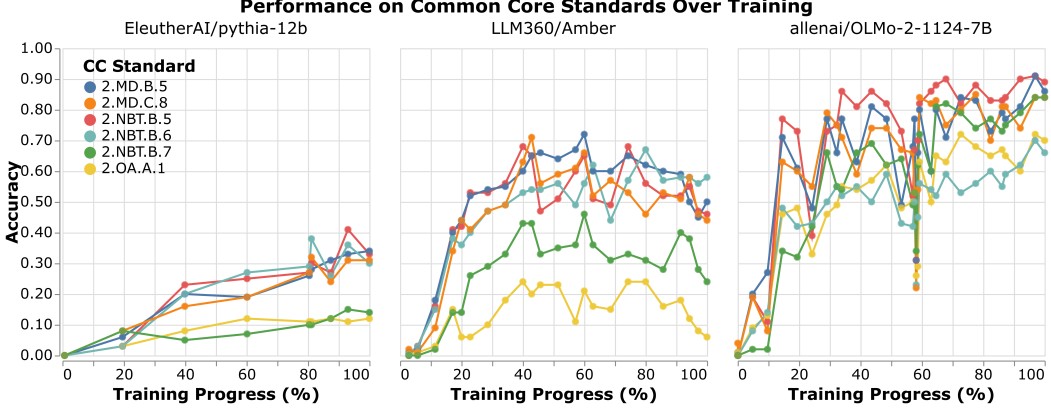

Figure 11: Learning Dynamics Across Amber, Pythia, OLMo for Second Grade

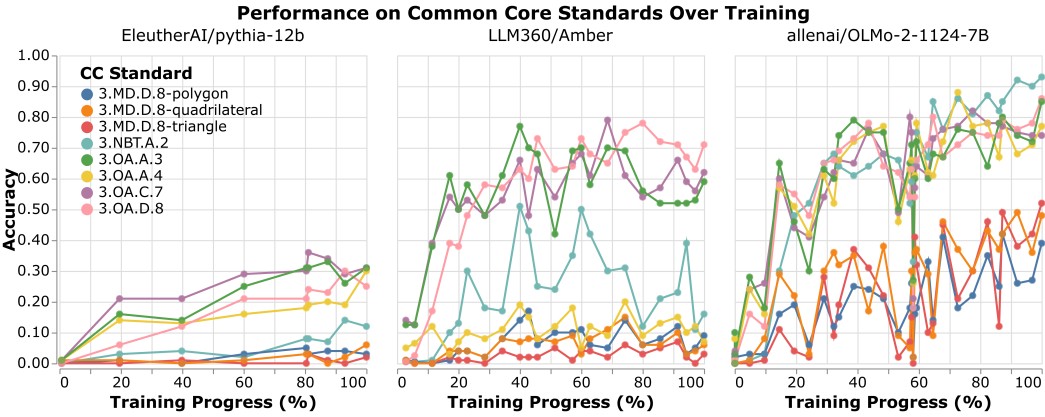

Figure 12: Learning Dynamics Across Amber, Pythia, OLMo for Third Grade

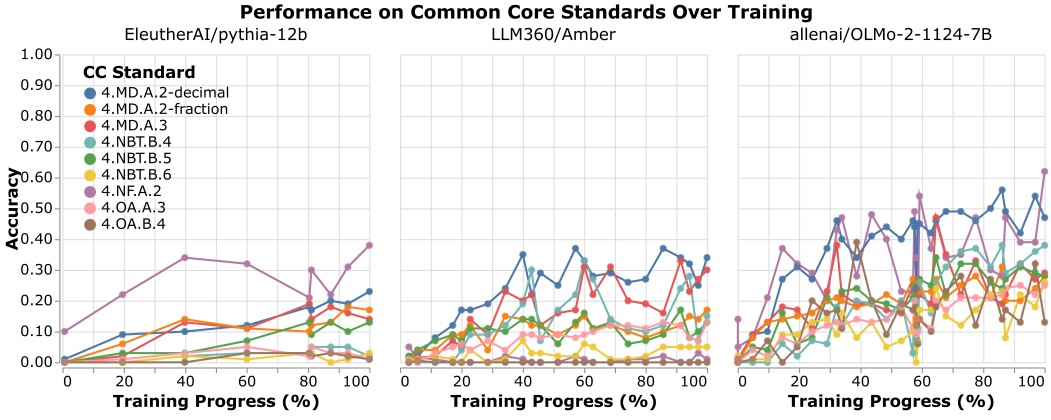

Figure 13: Learning Dynamics Across Amber, Pythia, OLMo for Fourth Grade

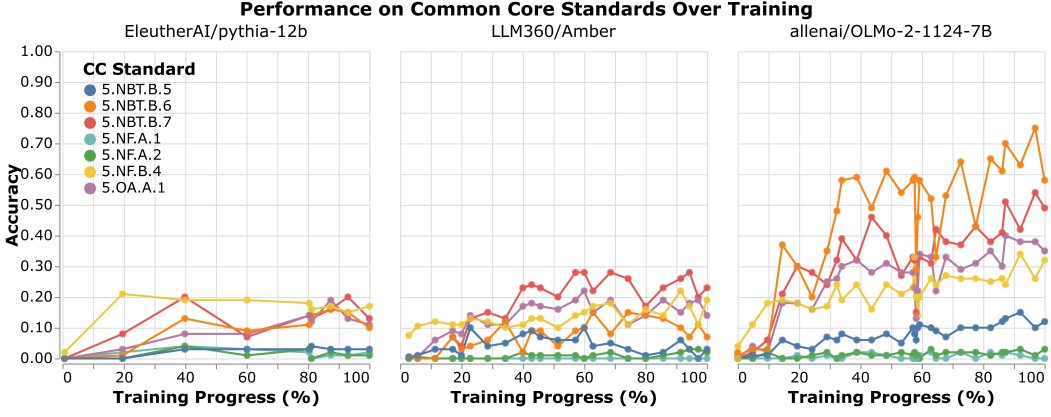

Figure 14: Learning Dynamics Across Amber, Pythia, OLMo for Fifth Grade

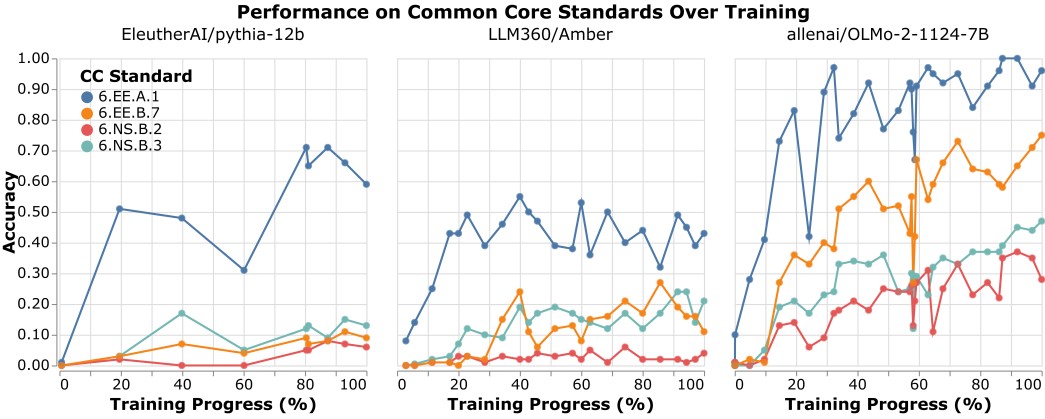

Figure 15: Learning Dynamics Across Amber, Pythia, OLMo for Sixth Grade

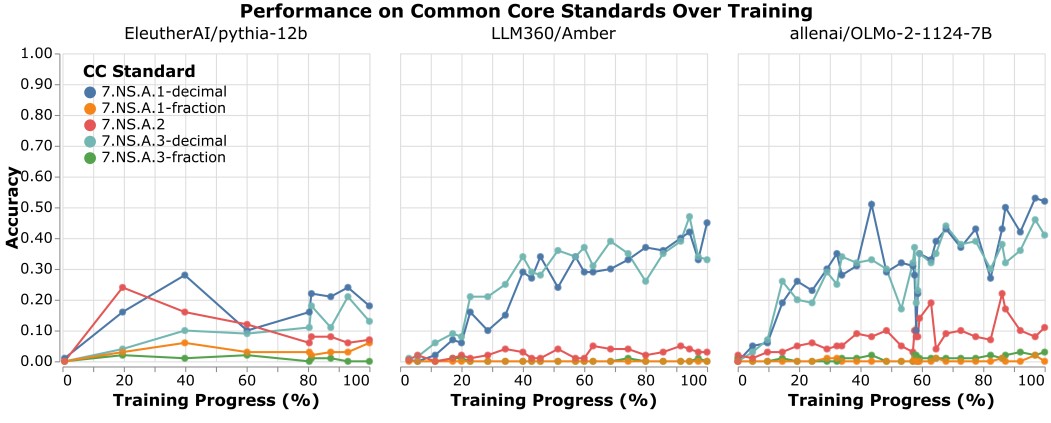

Figure 16: Learning Dynamics Across Amber, Pythia, OLMo for Seventh Grade

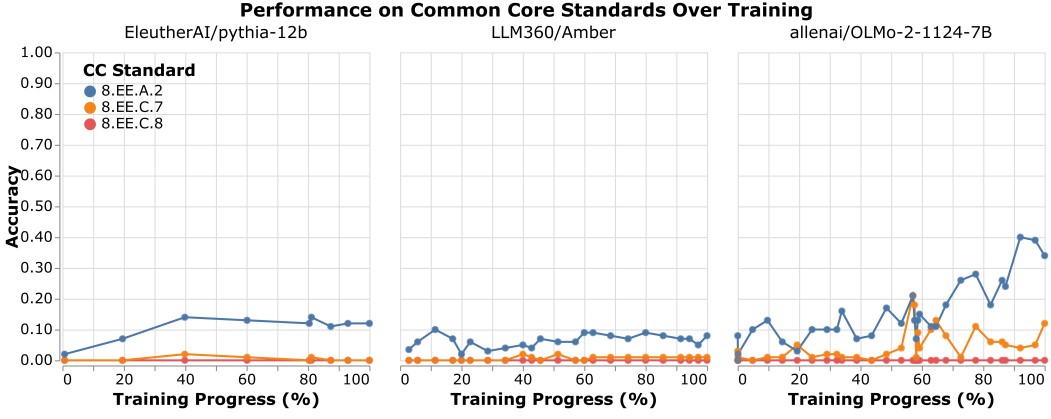

Figure 17: Learning Dynamics Across Amber, Pythia, OLMo for Eight Grade

| Standard ID | Description |
|---|---|
| 1.OA.A.1 | Use addition and subtraction within 20 to solve word problems involving situations of adding to, taking from, putting together, taking apart, and comparing, with unknowns in all positions, e.g., by using objects, drawings, and equations with a symbol for the unknown number to represent the problem. |
| 1.OA.A.2 | Solve word problems that call for addition of three whole numbers whose sum is less than or equal to 20, e.g., by using objects, drawings, and equations with a symbol for the unknown number to represent the problem. |
| 1.OA.D.8 | Determine the unknown whole number in an addition or subtraction equation relating three whole numbers. |

Table 2: CC Standards for Grade 1

| Standard ID | Description |
|---|---|
| 2.OA.A.1 | Use addition and subtraction within 100 to solve one- and two-step word problems involving situations of adding to, taking from, putting together, taking apart, and comparing, with unknowns in all positions, e.g., by using drawings and equations with a symbol for the unknown number to represent the problem. |
| 2.NBT.B.5 | Fluently add and subtract within 100 using strategies based on place value, properties of operations, and/or the relationship between addition and subtraction. |
| 2.NBT.B.6 | Add up to four two-digit numbers using strategies based on place value and properties of operations. |
| 2.NBT.B.7 | Add and subtract within 1000, using concrete models or drawings and strategies based on place value, properties of operations, and/or the relationship between addition and subtraction; relate the strategy to a written method. Understand that in adding or subtracting three-digit numbers, one adds or subtracts hundreds and hundreds, tens and tens, ones and ones; and sometimes it is necessary to compose or decompose tens or hundreds. |
| 2.MD.B.5 | Use addition and subtraction within 100 to solve word problems involving lengths that are given in the same units, e.g., by using drawings (such as drawings of rulers) and equations with a symbol for the unknown number to represent the problem. |
| 2.MD.C.8 | Solve word problems involving dollar bills, quarters, dimes, nickels, and pennies, using $ and ¢ symbols appropriately. |

Table 3: CC Standards for Grade 2

| Standard ID | Description |
|---|---|
| 3.OA.A.3 | Use multiplication and division within 100 to solve word problems in situations involving equal groups, arrays, and measurement quantities, e.g., by using drawings and equations with a symbol for the unknown number to represent the problem. |
| 3.OA.A.4 | Determine the unknown whole number in a multiplication or division equation relating three whole numbers. |
| 3.OA.C.7 | Fluently multiply and divide within 100, using strategies such as the relationship between multiplication and division (e.g., knowing that $8 \times 5 = 40$, one knows $40 \div 5 = 8$) or properties of operations. By the end of Grade 3, know from memory all products of two one-digit numbers. |
| 3.OA.D.8 | Solve two-step word problems using the four operations. Represent these problems using equations with a letter standing for the unknown quantity. Assess the reasonableness of answers using mental computation and estimation strategies including rounding. |
| 3.MD.D.8-triangle | Solve real world and mathematical problems involving perimeters of polygons, including finding the perimeter given the side lengths, finding an unknown side length, and exhibiting rectangles with the same perimeter and different areas or with the same area and different perimeters. |
| 3.MD.D.8-quadrilateral | Solve real world and mathematical problems involving perimeters of polygons, including finding the perimeter given the side lengths, finding an unknown side length, and exhibiting rectangles with the same perimeter and different areas or with the same area and different perimeters. |
| 3.MD.D.8-polygon | Solve real world and mathematical problems involving perimeters of polygons, including finding the perimeter given the side lengths, finding an unknown side length, and exhibiting rectangles with the same perimeter and different areas or with the same area and different perimeters. |
| 3.NBT.A.2 | Fluently add and subtract within 1000 using strategies and algorithms based on place value, properties of operations, and/or the relationship between addition and subtraction. |

Table 4: CC Standards for Grade 3

| Standard ID | Description |
|---|---|
| 4.OA.A.3 | Solve multistep word problems posed with whole numbers and having whole-number answers using the four operations, including problems in which remainders must be Interpreted. Represent these problems using equations with a letter standing for the unknown quantity. Assess the reasonableness of answers using mental computation and estimation strategies including rounding. |
| 4.OA.B.4 | Find all factor pairs for a whole number in the range 1-100. Recognize that a whole number is a multiple of each of its factors. Determine whether a given whole number in the range 1-100 is a multiple of a given one-digit number. Determine whether a given whole number in the range 1-100 is prime or composite. |
| 4.NBT.B.4 | Fluently add and subtract multi-digit whole numbers using the standard algorithm. |
| 4.NBT.B.5 | Multiply a whole number of up to four digits by a one-digit whole number, and multiply two two-digit numbers, using strategies based on place value and the properties of operations. Illustrate and explain the calculation by using equations, rectangular arrays, and/or area models. |
| 4.NBT.B.6 | Find whole-number quotients and remainders with up to four-digit dividends and one-digit divisors, using strategies based on place value, the properties of operations, and/or the relationship between multiplication and division. Illustrate and explain the calculation by using equations, rectangular arrays, and/or area models. |
| 4.NF.A.2 | Compare two fractions with different numerators and different denominators, e.g., by creating common denominators or numerators, or by comparing to a benchmark fraction such as 1/2. Recognize that comparisons are valid only when the two fractions refer to the same whole. Record the results of comparisons with symbols ¿, =, or ¡, and justify the conclusions, e.g., by using a visual fraction model. |
| 4.MD.A.2-decimal | Use the four operations to solve word problems involving distances, Intervals of time, liquid volumes, masses of objects, and money, including problems involving simple fractions or decimals, and problems that require expressing measurements given in a larger unit in terms of a smaller unit. Represent measurement quantities using diagrams such as number line diagrams that feature a measurement scale. |
| 4.MD.A.2-fraction | Use the four operations to solve word problems involving distances, Intervals of time, liquid volumes, masses of objects, and money, including problems involving simple fractions or decimals, and problems that require expressing measurements given in a larger unit in terms of a smaller unit. Represent measurement quantities using diagrams such as number line diagrams that feature a measurement scale. |
| 4.MD.A.3 | Apply the area and perimeter formulas for rectangles in real world and mathematical problems. |

Table 5: CC Standards for Grade 4

| Standard ID | Description |
|---|---|
| 5.OA.A.1 | Use parentheses, brackets, or braces in numerical expressions, and evaluate expressions with these symbols. |
| 5.NBT.B.5 | Fluently multiply multi-digit whole numbers using the standard algorithm. |
| 5.NBT.B.6 | Find whole-number quotients of whole numbers with up to four-digit dividends and two-digit divisors, using strategies based on place value, the properties of operations, and/or the relationship between multiplication and division. Illustrate and explain the calculation by using equations, rectangular arrays, and/or area models. |
| 5.NBT.B.7 | Add, subtract, multiply, and divide decimals to hundredths, using concrete models or drawings and strategies based on place value, properties of operations, and/or the relationship between addition and subtraction; relate the strategy to a written method and explain the reasoning used. |
| 5.NF.A.1 | Add and subtract fractions with unlike denominators (including mixed numbers) by replacing given fractions with equivalent fractions in such a way as to produce an equivalent sum or difference of fractions with like denominators. |
| 5.NF.A.2 | Solve word problems involving addition and subtraction of fractions referring to the same whole, including cases of unlike denominators, e.g., by using visual fraction models or equations to represent the problem. Use benchmark fractions and number sense of fractions to estimate mentally and assess the reasonableness of answers. |
| 5.NF.B.4 | Apply and extend previous understandings of multiplication to multiply a fraction or whole number by a fraction. |

Table 6: CC Standards for Grade 5

| Standard ID | Description |
|---|---|
| 6.NS.B.2 | Fluently divide multi-digit numbers using the standard algorithm. |
| 6.NS.B.3 | Add, subtract, multiply, and divide decimals to hundredths, using concrete models or drawings and strategies based on place value, properties of operations, and/or the relationship between addition and subtraction; relate the strategy to a written method and explain the reasoning used. |
| 6.EE.A.1 | Write and evaluate numerical expressions involving whole-number exponents. |
| 6.EE.B.7 | Solve real-world and mathematical problems by writing and solving equations of the form x + p = q and px = q for cases in which p, q and x are all nonnegative rational numbers. |

Table 7: CC Standards for Grade 6

| Standard ID | Description |
|---|---|
| 7.NS.A.1-fraction | Apply and extend previous understandings of addition and subtraction to add and subtract rational numbers; represent addition and subtraction on a horizontal or vertical number line diagram. |
| 7.NS.A.1-decimal | Apply and extend previous understandings of addition and subtraction to add and subtract rational numbers; represent addition and subtraction on a horizontal or vertical number line diagram. |
| 7.NS.A.2 | Apply and extend previous understandings of multiplication and division and of fractions to multiply and divide rational numbers. |
| 7.NS.A.3-fraction | Solve real-world and mathematical problems involving the four operations with rational numbers. |
| 7.NS.A.3-decimal | Solve real-world and mathematical problems involving the four operations with rational numbers. |

Table 8: CC Standards for Grade 7

| Standard ID | Description |
|---|---|
| 8.EE.A.2 | Use square root and cube root symbols to represent solutions to equations of the form $x^2 = p$ and $x^3 = p$, where p is a positive rational number. Evaluate square roots of small perfect squares and cube roots of small perfect cubes. Know that the square root of 2 is irrational. |
| 8.EE.C.7 | Solve linear equations in one variable. |
| 8.EE.C.8 | Analyze and solve pairs of simultaneous linear equations. |

Table 9: CC Standards for Grade 8

## D  Data generation pipeline details

### D.1  Grammar

We implemented a global attribute grammar in Python, where production rules are implemented as recursive Python functions. Effectively, each CC standard has its own grammar, composed of pieces from components from the global CC grammar, as well as possibly adding unique non-terminals. Each CC standard contains the following parameters:

**Description:** The description of the CC standard.

**Short description:** A shortened description of the CC standard.

**Filters:** A list of problem filters to ensure that all problems in this standard satisfy some requirement given in the Common Core description of the standard. The Problem-Length filter makes sure that the problem is within the desired length. CheckIntermediateValues filters out any problems with intermediate values greater or lesser than max_value or min_value, respectively. The ChainsOfVariables filter eliminates any problems where variables are assigned to equal exactly another variable, and nothing else. The ContainsTen filter checks if the math word problem contains numbers adding up to 10, or contains a 10 in the problem (for standards K.OA.A.4 and K.NBT.A.1, respectively).

**Transforms:** List of problem transformations applied to all symbolic structures from this standard. The NoUseles sVariables transform performs dead code elimination — it removes any variables that do not contribute to the final answer by applying a simple graph reachability algorithm on a dependency graph between statements, removing statements that the answer does not depend on. The Simplify transform essentially inlines variables that are used only once.

**Expressions:** Lists non-terminals available to generate expressions in symbolic structures for this standard. For example, this can make specific binary operations (e.g. addition, division) available on that particular standard.

**Min/max value:** Specifies bounds on values for both the final answer and all intermediate values in the solution.

**Min/max number:** Specifies bounds on numeric constants sampled in the symbolic structure.

**Max depth:** Sets a maximum depth for expressions in the symbolic structure.

**Samples:** We include 2+ hand-written, standard-relevant examples of a symbolic problem followed by a relevant natural language problem generation, which we use as few-shot prompts during problem generation. We also use these prompts, but in reverse (natural language followed by symbolic problem), when we prompt GPT-4 during cycle consistency.

### D.2 Answer Grading During Evaluation

Given a solution in natural language, we first use a rule-based answer extractor to extract any model's numerical answer. In cases where a language model doesn't answer in the required format, or answers in an unexpected format, the answer is initially marked as incorrect. For all problems with incorrect answers, we use Llama-3 70B to re-extract the final answer. We few-shot prompt it with hand-generated examples of solutions and extracted final answers, and ask it to extract the final answer from the new solution. If a problem that was previously incorrect is marked as correct (given the newly extracted answer), we rerun the model on any followups the problem might have. Note that this "regrading" step can only improve accuracy from the base result, since we only run it on problems that failed under the rule-based evaluation. In practice, we found this process to have negligible false-positive rate — only in a handful of cases across all models we observed either answer extraction processes extracting the correct answer out of a wrong response (e.g., if the answer to a problem is 2, and the model responds "On day 2, Sally bought 9 dolls", the rule-based parser extracts 2 as being the model's answer, though the sentence implies its answer to be 9). On the other hand, the LLaMA-3 70B extractor greatly reduces our false negative rate in a handful of models (especially DeepSeek) which are more likely to respond in a format different from what our prompt asks for.

### D.3 Cost estimate

All problems in MathCAMPS were generated using OpenAI `gpt-4-0613`, in May 2024. We estimate an approximate cost of 330 USD to generate 9607 problems (including main problems and follow-ups). This includes the cost to perform cycle consistency, and problems that are discarded by cycle consistency. This gives an average cost of 0.034 USD (3.4 cents) per cycle-consistent problem or follow-up question.

## E  Correlation between MathCAMPS and GSM8k

Figure 18 shows accuracies of several models on both GSM8k and MathCAMPS, along with the line of best fit. There is a strong correlation between overall accuracy in both datasets ($\rho = 0.91$, $p < 10^{-6}$), though MathCAMPS allows for many fine-grained analysis besides overall performance.

## F  Cycle consistency efficacy and failure cases

This cycle consistency test significantly improves the reliability of our pipeline. We manually evaluated 245 random problems generated by sampling a symbolic structure and then a word problem from GPT-4. Out of those, we identified 30 word problems (12.2%) that were not faithful to the original symbolic structure — for those, the answer that we compute to the *symbolic problem* does not match our manual solution to the *word problem*. Cycle consistency discarded 25 of those (and 7 problems that were indeed faithful). Out of the remaining 215 problems, 210 (97.7%) were judged as faithful in our manual check. The results are shown in Table 10.

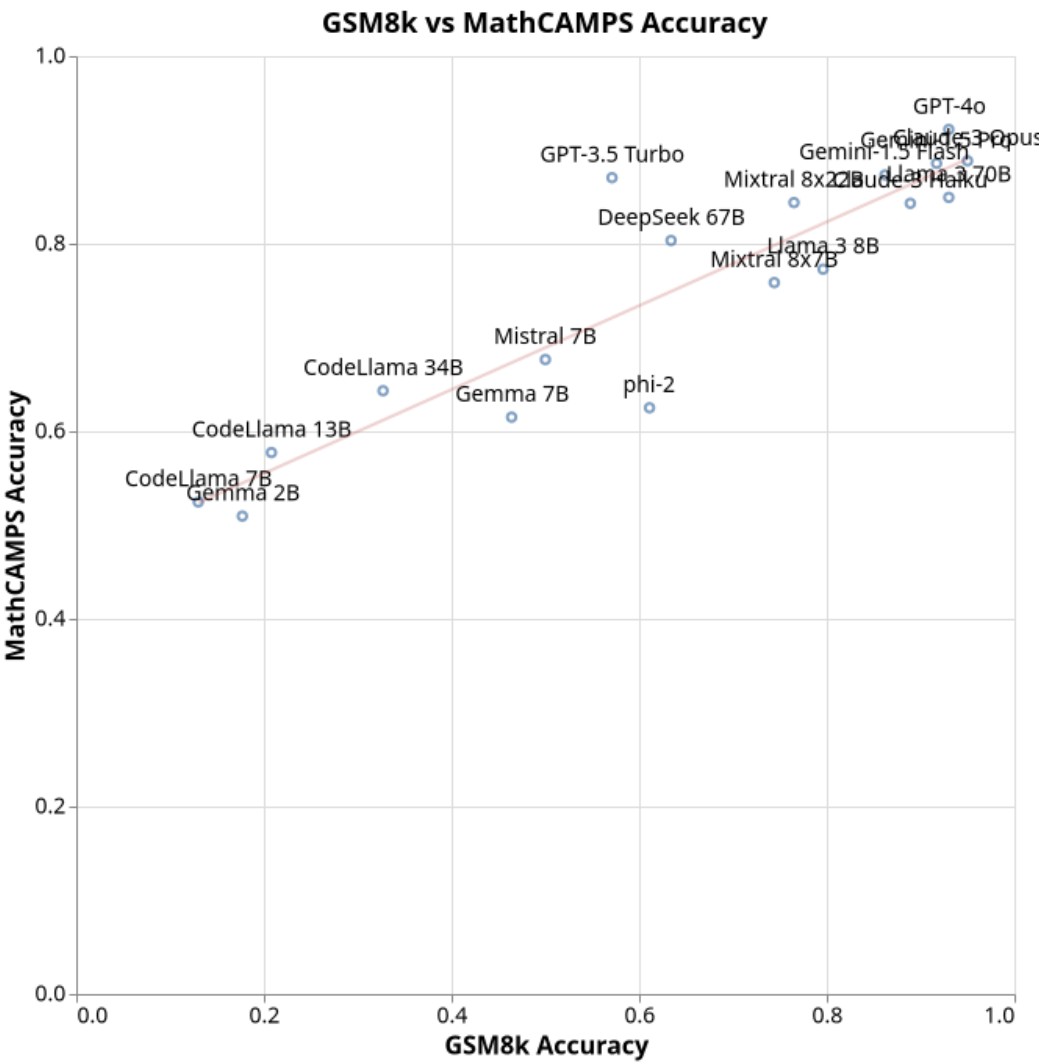

Figure 18: Correlation between accuracy on GSM8k and on MathCAMPS ($\rho = 0.91$).

|  | Faithful problem | Unfaithful problem |
|---|---|---|
| Cycle-consistent | 208 | 5 |
| Not cycle-consistent | 7 | 25 |

Table 10: Efficacy of Cycle Consistency

### F.1 Unfaithful but cycle-consistent problems

We noted that of the five problems that were unfaithful but cycle-consistent, two had ambiguous language that led to the generation of a cycle-consistent symbolic structure. The three other problems were all from the same CC standard (4.NBT.B.6, a skill regarding the calculation of quotients and remainders).

### F.2 Unfaithful and non-cycle consistent problems

Below, we provide threeexamples of unfaithful problem generations that were unfaithful and correctly discarded by the cycle-consistency process.

**Original symbolic structure:** [[var f = (5 - 4)]][[var t = (4 - f)]][[question d = ['t']]] theme: Rocket

**Generated word problem:** In the space exploration research center, Diego has 5 rockets. He launched 1 rocket for testing purposes. After this, he orders 4 new rockets, but doesn't count the one that he just launched. How many rockets does Diego have now?

**New symbolic structure:** [[var initial_rockets = 5]][[var launched_rockets = 1]][[var ordered_rockets = 4]][[var total_rockets = (initial_rockets - launched_rockets + ordered_rockets)]][[question result = ['total_rockets']]]

**Original symbolic structure:** [[var v = (79.24 * 37.6)]][[question s = ['v']]] theme: Treasure chest

**Generated word problem:** A pirate finds a treasure chest full of golden coins. Each golden coin weighs 79.24 grams. If the total weight of the coins is 37.6 kilograms, how many golden coins are there in the treasure chest?

**New symbolic structure:** [[var weightInGrams = (37.6 * 1000)]][[var n = (weightInGrams / 79.24)]][[question numCoins = ['n']]]

**Original symbolic structure:** [[var j = (4450 / 89)]][[var c = ((69 * j) + j)]][[question b = ['c']]] theme: Sandwich

**Generated word problem:** The Sandwich Factory makes sandwiches at a rate of 4450 sandwiches every 89 days. If the factory not only produces 69 sandwiches each day, but also an additional batch of the exact same rate each day, how many sandwiches are produced in total each day?

**New symbolic structure:** [[var a = 4450]][[var b = 89]][[var c = 69]][[var d = a / b]][[var e = c + d]][[question f = ['e']]]

### F.3 Faithful but non-cycle-consistent problems

The 7 faithful problems were discarded were because one of two issues. 6 our of 7 problems were discarded because their back-translated symbolic structure had a small structural error (e.g. and extra square bracket at the end of a variable declaration). The 7th problem had a genuine error in its back translation, which caused the new symbolic structure to have a different final answer than the original symbolic structure, causing us to discard the problem.

## G Familiarity bias

MathCAMPS was generated using GPT-4. GPT-4o, a model of the same family, was also the best performer overall (Table H). To test whether this might be due to a familiarity bias — problems being in-distribution for GPT-4o, but out-of-distribution for other models —, we generated a 10%-scale dataset using the exact same pipeline, but using Claude 3 Opus for both generating word problems and testing cycle consistency. This dataset has the same distribution of standards as MathCAMPS. We evaluated GPT-4o and Claude 3 Opus on this dataset — accuracies are reported in Table 11. GPT-4o also performs better in this dataset,

| Model | GPT4-generated MathCAMPS accuracy | Claude-generated MathCAMPS accuracy |
|---|---|---|
| GPT-4o | 0.910 | 0.954 |
| Claude 3 Opus | 0.887 | 0.909 |

Table 11: Performance of GPT-4o and Claude 3 Opus on the dataset genreated using Claude

| Vendor | Model | All | K | 1 | 2 | 3 | 4 | 5 | 6 | 7 | 8 |
|---|---|---|---|---|---|---|---|---|---|---|---|
| OpenAI | GPT-4o | 0.92 | 0.98 | 0.98 | 0.98 | 0.98 | 0.92 | 0.88 | 0.95 | 0.89 | 0.64 |
| Anthropic | Claude-3 Opus | 0.89 | 0.97 | 0.99 | 0.96 | 0.98 | 0.89 | 0.83 | 0.96 | 0.73 | 0.56 |
| Google | Gemini-1.5 Pro | 0.89 | 0.95 | 0.98 | 0.97 | 0.97 | 0.89 | 0.83 | 0.93 | 0.78 | 0.54 |
| Google | Gemini-1.5 Flash | 0.87 | 0.98 | 0.98 | 0.97 | 0.98 | 0.80 | 0.80 | 0.90 | 0.84 | 0.56 |
| OpenAI | GPT-3.5 Turbo | 0.87 | 0.96 | 0.98 | 0.98 | 0.97 | 0.86 | 0.77 | 0.90 | 0.77 | 0.56 |
| Anthropic | Claude-3 Sonnet | 0.86 | 0.96 | 0.98 | 0.97 | 0.98 | 0.88 | 0.74 | 0.94 | 0.66 | 0.49 |
| Anthropic | Claude-3 Haiku | 0.84 | 0.97 | 0.98 | 0.97 | 0.98 | 0.87 | 0.69 | 0.92 | 0.59 | 0.51 |
| Qwen | Qwen2-Math 72B | 0.89 | 0.98 | 0.99 | 0.98 | 0.97 | 0.90 | 0.80 | 0.91 | 0.77 | 0.59 |
| Meta | Llama 3 70B | 0.85 | 0.96 | 0.97 | 0.97 | 0.97 | 0.85 | 0.71 | 0.87 | 0.73 | 0.50 |
| Mistral | Mixtral 8x22B | 0.84 | 0.96 | 0.99 | 0.98 | 0.96 | 0.79 | 0.69 | 0.88 | 0.73 | 0.61 |
| Qwen | Qwen2-Math 7B | 0.83 | 0.96 | 0.99 | 0.97 | 0.93 | 0.85 | 0.66 | 0.91 | 0.58 | 0.62 |
| DeepSeek | DeepSeek 67B | 0.80 | 0.95 | 0.99 | 0.96 | 0.93 | 0.82 | 0.60 | 0.84 | 0.61 | 0.47 |
| DeepSeek | DeepSeek Math 7B Base | 0.78 | 0.94 | 0.97 | 0.93 | 0.89 | 0.75 | 0.63 | 0.86 | 0.53 | 0.55 |
| Numina | NuminaMath 7B TIR | 0.78 | 0.89 | 0.97 | 0.95 | 0.90 | 0.72 | 0.63 | 0.84 | 0.59 | 0.53 |
| Meta | Llama 3 8B | 0.77 | 0.94 | 0.97 | 0.96 | 0.94 | 0.78 | 0.55 | 0.79 | 0.53 | 0.43 |
| Mistral | Mixtral 8x7B | 0.76 | 0.94 | 0.96 | 0.93 | 0.91 | 0.75 | 0.52 | 0.80 | 0.53 | 0.45 |
| InternLM | InternLM-Math Base 20B | 0.74 | 0.95 | 0.96 | 0.95 | 0.86 | 0.68 | 0.55 | 0.79 | 0.52 | 0.47 |
| EleutherAI | Llemma 34B | 0.71 | 0.95 | 0.96 | 0.93 | 0.87 | 0.61 | 0.47 | 0.77 | 0.46 | 0.44 |
| Mistral | Mistral 7B | 0.68 | 0.89 | 0.94 | 0.91 | 0.84 | 0.61 | 0.42 | 0.66 | 0.45 | 0.42 |
| DeepSeek | DeepSeek Coder 33B | 0.65 | 0.88 | 0.93 | 0.92 | 0.83 | 0.54 | 0.36 | 0.66 | 0.44 | 0.38 |
| Meta | CodeLlama 34B | 0.64 | 0.90 | 0.94 | 0.92 | 0.85 | 0.51 | 0.38 | 0.70 | 0.37 | 0.30 |
| Microsoft | phi-2 | 0.63 | 0.95 | 0.96 | 0.89 | 0.78 | 0.46 | 0.38 | 0.61 | 0.37 | 0.41 |
| EleutherAI | Llemma 7B | 0.62 | 0.78 | 0.90 | 0.85 | 0.79 | 0.48 | 0.41 | 0.67 | 0.41 | 0.36 |
| Google | Gemma 7B | 0.62 | 0.83 | 0.92 | 0.90 | 0.82 | 0.47 | 0.36 | 0.65 | 0.36 | 0.30 |
| Meta | CodeLlama 13B | 0.58 | 0.87 | 0.92 | 0.87 | 0.75 | 0.41 | 0.30 | 0.61 | 0.32 | 0.34 |
| InternLM | InternLM-Math Base 7B | 0.58 | 0.71 | 0.73 | 0.73 | 0.72 | 0.54 | 0.38 | 0.61 | 0.37 | 0.39 |
| Meta | CodeLlama 7B | 0.52 | 0.85 | 0.92 | 0.84 | 0.69 | 0.37 | 0.25 | 0.57 | 0.25 | 0.16 |
| Google | Gemma 2B | 0.51 | 0.66 | 0.76 | 0.74 | 0.67 | 0.42 | 0.28 | 0.55 | 0.30 | 0.27 |
| - | Avg. Performance | 0.75 | 0.91 | 0.95 | 0.92 | 0.88 | 0.70 | 0.57 | 0.79 | 0.56 | 0.46 |

Table 12: Final answer accuracy of LLMs on MathCAMPS, both over all problems (**All**) and considering only standards in each grade we cover (**K to 8**). Highlights compare to gradewise avg.

suggesting that its performance in MathCAMPS was not due to a higher relative familiarity with the problems.

# H   Performance of Families of Models on MathCAMPS

Table H shows both aggregate accuracy on MathCAMPS, as well as accuracy across standards partitioned by grade, whereas Figure 18 compares the aggregate accuracies on Math-CAMPS and GSM8K. Closed-weights models are shown above the line, with open-weights models below. GPT-4o ranks at the top in overall accuracy. Since we used GPT-4 to generate the problems, we must rule out familiarity bias (Stureborg et al., 2024) in this result. We thus generated a 10%-scale dataset with the same pipeline but using Claude-3 Opus. We found

| Model | Top outlier skill | Rank change |
|---|---|---|
| GPT-4o | 8.EE.C.8 - Solve two-variable systems | $(1^{st} \searrow 22^{th})$ |
| Claude-3 Opus | 2.MD.B.5 - Add/sub within 100 | $(2^{nd} \searrow 18^{th})$ |
| Gemini-1.5 Pro | K.OA.A.4 - Adding to equal 10 | $(4^{th} \searrow 23^{th})$ |
| Claude-3 Haiku | 6.EE.A.1 - Evaluate exponents | $(10^{th} \searrow 20^{th})$ |
| Llama 3 70B | 3.OA.A.3 - Mul/div within 100 | $(8^{th} \searrow 21^{th})$ |
| Mixtral 8x22B | 8.EE.C.8 - Solve two-variable systems | $(9^{th} \searrow 21^{th})$ |
| Qwen2-Math 7B | 8.EE.C.8 - Solve two-variable systems | $(11^{th} \searrow 25^{th})$ |
| DeepSeek 67B | K.NBT.A.1 - Decompose into 10s | $(12^{th} \nearrow 1^{st})$ |
| Llama 3 8B | K.OA.A.4 - Adding to equal 10 | $(15^{th} \nearrow 3^{rd})$ |
| Mixtral 8x7B | 6.EE.A.1 - Evaluate exponents | $(16^{th} \searrow 26^{th})$ |
| InternLM-Math Base 20B | 2.NBT.B.5 - Add/sub within 100 | $(17^{th} \nearrow 2^{nd})$ |
| Llemma 34B | 3.OA.A.3 - Mul/div within 100 | $(18^{th} \nearrow 1^{st})$ |
| Mistral 7B | 1.OA.A.1 - Add/sub within 20 | $(19^{th} \searrow 26^{th})$ |
| DeepSeek Coder 33B | 6.EE.A.1 - Evaluate exponents | $(20^{th} \nearrow 3^{rd})$ |
| phi-2 | K.OA.A.4 - Adding to equal 10 | $(22^{th} \nearrow 4^{th})$ |
| Llemma 7B | 6.EE.A.1 - Evaluate exponents | $(23^{th} \nearrow 5^{th})$ |
| Gemma 7B | K.OA.A.5 - Add/sub within 5 | $(24^{th} \nearrow 6^{th})$ |
| InternLM-Math Base 7B | 4.OA.B.4 - Factor pairs within 100 | $(26^{th} \nearrow 15^{th})$ |
| CodeLlama 7B | 8.EE.C.8 - Solve two-variable systems | $(27^{th} \nearrow 15^{th})$ |
| Gemma 2B | 8.EE.C.8 - Solve two-variable systems | $(28^{th} \nearrow 11^{th})$ |

Table 13: Largest model rank changes when focusing on one CC standard. Here, A $\nearrow$B indicates that the model ranks A$^{th}$ on MathCAMPS overall, but ranks B$^{th}$ when only evaluating on problems from the indicated CC standard. Conversely, $\searrow$marks notable cases where a model's performance on the indicated CC standard is lower than its overall performance on MathCAMPS. We show selected rows here, the complete table can be found in the Appendix.

that GPT-4o still outperforms Claude-3 Opus on this dataset (see Appendix G), suggesting that its advantage on MathCAMPS was not due to a familiarity bias.

We make the following observations:

**Models of similar overall performance can have large disparities in specific abilities or grades.** Several models that have comparable overall accuracies show large differences when compared on specific mathematical skills. As an example, Claude-3 Opus and Claude-3 Sonnet have similar overall accuracy both in MathCAMPS (.89 vs .86) and in GSM8K (.95 vs .923). However, we find that Claude-3 Opus is significantly better at manipulating fractions. For instance, in the CC standard 5.NF.A.2, described as *"Solve word problems involving addition and subtraction of fractions referring to the same whole, including cases of unlike denominators"*, Opus has a 36% advantage over Sonnet, scoring a 70% accuracy for this standard, whereas Sonnet only achieves 34%. Similarly, while Gemma 7B and phi-2 have comparable overall performance (.62 vs .63 accuracy on MathCAMPS), some capabilities in each model seem nearly absent from the other. Gemma 7B is highly accurate when performing multi-digit multiplication — an ability stressed in standard 4.NBT.B.4, where Gemma 7B achieves 94% accuracy. In stark contrast, phi-2 only solves 22% of those problems. On the other direction, phi-2 is one of the highest performing models on 4.NF.A.2 ("Compare two fractions with different numerators and different denominators"), with 90% accuracy. In this same standard, Gemma 7B only scores 19%. Such stark differences are obscured when only analyzing aggregate metrics, whereas MathCAMPS allows for a much more nuanced understanding of mathematical reasoning capabilities.

**Overall ranking between models is largely a function of which skills we choose to evaluate.** Overall accuracies in any dataset induce a single performance ranking of models. However, when we look at individual CC standards in MathCAMPS, rankings are largely a function of which skills we choose to evaluate. Comparing pairs of models across all standards, rarely we find cases where one model Pareto-dominates another (i.e. is better on all standards): only 23.08% of all pairs of models have a Pareto winner. Table H shows how the ranking of a model in individual skills can often deviate strongly from its overall ranking. Here, the first ordinal in each cell shows the model's global ranking when comparing overall performance in MathCAMPS, whereas the second shows the model's ranking on that particular CC standard. We find many cases of large discrepancies. For instance, on systems of equations, GPT-4o tends to excessively rely on decimal approximations when operating with fractions, resulting in poor performance. Llemma 34B, which places 13th overall, is the best performing model on a simple kindergarten-level word problems on adding to complete 10.

**Aggregate accuracies are strongly correlated between GSM8k and MathCAMPS** When considering overall performance, the trends in GSM8k hold on the novel problems from MathCAMPS, which cover overlapping topics (Pearson correlation of 0.865, $p < 10^{-5}$; we show this correlation in Figure 18). This correlation corroborates the progress that public benchmarks have witnessed, suggesting that data contamination does not play a major role in explaining observed improvements in recent LLMs. We note that prior work attempting to replicate the distribution of GSM8k, such as the independent effort to collect GSM1k (Zhang et al., 2024), has observed a smaller correlation, including substantial drops in performance for some models. This is entirely compatible with our findings here, due to the difficulty of exactly replicating the distribution over skills in any given human-created benchmark. As the sharp differences in Table H indicate, an (unintended) shift in this distibution can drastically — and unevenly — affect accuracy, even if no data contamination occurs. These shifts are easily avoided in an automated pipeline as in MathCAMPS, allowing us to draw new problems from the exact same distribution in the future.

## H.1 Standard-specific analysis

Despite decently high performance across the board, GPT-4o's performance fell at or below 90% on the following skills: 4.MD.A.2-fraction, 4.OA.A.3, 5.NF.A.1, 7.NS.A.3-fraction, and 8.EE.C.8. At their core, all these abilities require fraction addition or subtraction, a skill we noted that GPT-4o struggles with. Specifically, the model starts approximating fractions using decimals, and the error introduced by this compounds throughout the problem, resulting in an incorrect final answer. Surprisingly, GPT-4o achieves an 86% on 5.NF.B.4, which requires fraction multiplication, indicating that it is likely the multi-step process of finding common denominators in adding/subtracting fractions that challenges GPT-4o. Additionally, GPT-4o achieves performances above 90% on 4.MD.A.2-decimal and 7.NS.A.3-decimal, which are the CC standards equivalent to 4.MD.A.2-fraction and 7.NS.A.3-fraction, using decimals instead of fractions in the problems. This trend isn't isolated to the GPT models, though, as most models tended to struggle more with standards involving fractions.

Work from Lucy et al. (2024) showed that over 50% of problems from GSM8K originated from three CC standards, namely, 4.OA.A.3 (20.73%), 2.OA.A.1 (16.58%), and 3.OA.D.8 (15.75%). These standards ask students to solve multistep word problems involving the four operations, use addition and subtraction to solve two-step word problems within 100, and solve two-step word problems using the four operations, respectively. While most models we experimented with performed relatively well on 2.OA.A.1 and 3.OA.D.8, CC standard 4.OA.A.3 did prove to be challenging, with the most performant model, Qwen2-Math 72B, achieving an 86% on the standard.

Out of the 49 total skills we evaluated (44 standards, some of which we split into sub-standards), 19 skills had an absolute winner: a model which outperforms all other models on that skill. The distribution of these skills is given in Table 14. This analysis shows that even generally weaker models, such as GPT-3.5 Turbo, have particular skills that they excel

| Model | Standards Won |
|---|---|
| GPT-4o | 4.NBT.B.6, 7.NS.A.2, 8.EE.C.7, 7.NS.A.1-fraction, 5.NF.A.1, 7.NS.A.3-fraction |
| Qwen2-Math 72B | 1.OA.A.1, 3.OA.D.8, 5.NF.B.4, 4.OA.A.3, 4.MD.A.2-fraction |
| GPT-3.5 Turbo | 2.NBT.B.6, 5.OA.A.1, 8.EE.C.8 |
| Claude-3 Opus | 6.NS.B.2, 5.NBT.B.7 |
| Gemini-1.5 Flash | 7.NS.A.3-decimal, 5.NF.A.2 |
| Claude-3 Sonnet | 3.MD.D.8-polygon |

Table 14: Standards with strict winners, i.e., models who strictly outperform all other models on that standard.

| Model | Acc. w.f. | Largest accuracy drop w/ follow-ups | |
|---|---|---|---|
| GPT-4o | 0.82 | 5.NF.A.1 - Add/sub fractions | 0.86 ↘0.58) |
| Claude-3 Opus | 0.76 | 7.NS.A.1-fraction - Add/sub with fractions | 0.54 ↘0.23) |
| Gemini-1.5 Pro | 0.77 | 5.OA.A.1 - Evaluating with parentheses | 0.95 ↘0.69) |
| Claude-3 Haiku | 0.70 | 7.NS.A.2 - Mult/div with fractions | 0.55 ↘0.26) |
| Qwen2-Math 72B | 0.78 | 5.NF.A.1 - Add/sub fractions | 0.49 ↘0.23) |
| Llama 3 70B | 0.69 | 4.NF.A.2 - Compare two fractions | 0.99 ↘0.66) |
| Mixtral 8x22B | 0.69 | 7.NS.A.1-fraction - Add/sub with fractions | 0.69 ↘0.17) |
| Qwen2-Math 7B | 0.71 | 5.NF.A.2 - Add/sub fraction word problems | 0.41 ↘0.17) |
| DeepSeek Math 7B Base | 0.65 | 5.NF.B.4 - Mult fractions | 0.81 ↘0.57) |
| NuminaMath 7B TIR | 0.62 | 5.NF.A.2 - Add/sub fraction word problems | 0.44 ↘0.18) |
| Llama 3 8B | 0.58 | 4.NF.A.2 - Compare two fractions | 0.90 ↘0.52) |
| Mixtral 8x7B | 0.58 | 7.NS.A.2 - Mult/div with fractions | 0.60 ↘0.28) |
| Llemma 34B | 0.55 | 5.NF.B.4 - Mult fractions | 0.68 ↘0.31) |
| Mistral 7B | 0.48 | 7.NS.A.1-decimal - Add/sub with decimals | 0.91 ↘0.50) |
| DeepSeek Coder 33B | 0.60 | 3.OA.A.3 - Mul/div within 100 | 0.95 ↘0.81) |
| phi-2 | 0.39 | 3.NBT.A.2 - Add/sub within 1000 | 0.71 ↘0.23) |
| Llemma 7B | 0.42 | 5.NF.B.4 - Mult fractions | 0.58 ↘0.21) |
| Gemma 7B | 0.33 | 7.NS.A.1-decimal - Add/sub with decimals | 0.91 ↘0.32) |
| InternLM-Math Base 7B | 0.42 | 7.NS.A.1-decimal - Add/sub with decimals | 0.82 ↘0.47) |
| CodeLlama 7B | 0.49 | 2.NBT.B.7 - Add/sub within 100 | 0.80 ↘0.67) |
| Gemma 2B | 0.24 | 3.NBT.A.2 - Add/sub within 1000 | 0.93 ↘0.26) |

Table 15: Model performance on our mathematical dialogue task, where the model must answer follow-up questions besides the initial problem. The second column, **Accuracy with follow-ups**, shows overall success rate across standards that contain follow-up questions, considering a model successful only when it answers a problem and its follow-up questions correctly. The third and fourth columns show the hardest standard for each model when it comes to follow-up questions, showing a standard's code and abbreviated description, the model's accuracy ignoring follow-ups, and after follow-ups. We show selected rows here, the complete table can be found in the Appendix.

on. This fact *is hidden when looking at aggregate accuracies*, but is revealed in our finer-grained analysis.

## H.2 Follow-up tasks

We now evaluate the performance of language models when asked follow-up questions. Here, we first give the initial problem, and in case the model answers correctly we ask either an incremental follow-up, a counterfactual follow-up, or both (in separate contexts), depending on the standard (some standards don't have follow-ups, and for some problems we failed to find a cycle-consistent follow-up within the max attempts). Here, we're interested in analyzing the (lack of) robustness that LMs might have when probed with extra questions — our follow-ups are generally answerable using the same core mathematical knowledge involved in the initial problem but require longer range attention and dialog understanding.

Table 15 shows overall accuracies when we only consider a model successful on a problem when it also answers its follow-up questions correctly. We also show the major accuracy drops across CC standards for each model (last two columns). We find many notable cases, in both stronger and weaker models. GPT-4o, for instance, is 90% accurate in evaluating expressions of addition of fractions with multi-digit numerators and denominators (5.NF.A.1 — notably, this requires putting fractions in the same denominator). When asked to add another fraction to the result, or change one of the original fractions to a new one and re-do the computation, its success rate when evaluated at correctly answering both follow-ups drops to 61%, or a 29% decrease. Other models drop even more dramatically. For instance, phi-2 solves 57% of the problems in 7.NS.A.2, which are about multiplying two fractions (only requires two multi-digit multiplications — we do not require the result to be in lowest terms). However, when asked to multiply the result by a further third fraction, phi-2 tends to not reuse its previous (correct) result, and instead proceeds by writing down the product of the three numerators (and denominators), and attempt to directly evaluate this product. This strategy is rarely successful, and it only achieves 8% accuracy when accounting for the follow-ups (an absolute 49% drop). Overall, we find many cases where models are not robust to simple follow-up questions. We hypothesize that this setup of mathematical dialogue is much less frequent in pre-training data, and that follow-up problems in MathCAMPS can be a rich source of further analyses for future work.

