# OpenReview forum: "From Next-Token to Mathematics: The Learning Dynamics of Mathematical Reasoning in Language Models"
_colmweb.org/COLM/2025/Conference — COLM 2025_

### Official Review · Reviewer_Mhg7 · 2025-05-11

**Rating:** 7
**Confidence:** 3
**Ethics Flag:** 1

**Summary:**

This paper proposed to study the acquisition of math abilities during pretraining and instruction tuning. The authors first define a problem generation procedure that synthesize problems from common core objectives. Then, the authors studied the performance of models in different checkpoints of pre-training on these generated problems. Last, the authors studied the performance change between pretraining and instruction finetuning. Overall, the authors found that the math skill acquisition is genenerally monotonic and the model learns to solve problems of different difficulty simultaneously (rather than one by one). Furthermore, instruction finetuning has different effect on different models.

**Reasons To Accept:**

The paper's investigation is new and the paper is also well written.

The data generation procedure is quite novel and could be used to synthesize even harder questions.

The analysis is generally comprehensive, with intuitive results.

**Reasons To Reject:**

I don't think this paper has critical weaknesses, but a few things that could be improved:

1. Compared to pretraining, the finetuning analysis feels a bit rushed, with only the original and final checkpoints before and after finetuning. As finetuning is much lighter than pretraining, the authors may be able to conduct custom finetuning to even strengthen the paper's analysis.

2. The authors should do an ablation study on the quality of the data generation compared to directly prompting GPT-4o to generate problems on a given theme according to the common core description. It is unclear if we actually need all the complexity in the current data generation pipeline.

3. The figures should be made in vector graphics. Right now, many plots are pixelated when zoomed in. Also, I don't recommend the use of bold font in plot as they are too distractive on the normal flow of reading.

---

> ### Author Response · Authors · 2025-05-31
>
> We thank the reviewer for their attention to detail with all the feedback! We have addressed the comments below:
>
> > Compared to pretraining, the finetuning analysis feels a bit rushed, with only the original and final checkpoints before and after finetuning. As finetuning is much lighter than pretraining, the authors may be able to conduct custom finetuning to even strengthen the paper's analysis.
>
> Thank you for suggesting improvements to the fine-tuning analysis! We agree that controlled experiments isolating the effects of the instruction-tuning data corpus would be useful in isolating how various factors within the data impact the final performance of the model. While we were not able to conduct such experiments due to the high computational cost of fine-tuning LLMs, our current works take a first step by analyzing the impact of post-training on three publicly available models (Amber, Llama, and OLMo), each of which underwent a distinct post-training schema. This cross-model comparison highlights general trends and introduces hypotheses we hope future works can explore in more controlled manners.
>
> Our results are also in alignment with an observation made by the very recent work of Springer et. al. [1], which considers the quantity of data seen during the pre-training of the LLM and its instruction-tuned counterpart’s performance. They show that the more tokens an LLM sees during pre-training, the less effective instruction-tuning is on it. In our case, OLMo and Llama, which each saw 15T and 4T tokens of data during pre-training respectively, saw more degradation post fine tuning compared to Amber, which saw ~1.3T tokens during pre-training. We will add a brief discussion of Springer et. al.’s paper as a possible explanation in our final manuscript.
>
> [1] J. M. Springer et. al. Overtrained Language Models Are Harder to Fine-Tune (https://arxiv.org/abs/2503.19206)
>
> > The authors should do an ablation study on the quality of the data generation compared to directly prompting GPT-4o to generate problems on a given theme according to the common core description. It is unclear if we actually need all the complexity in the current data generation pipeline.
>
> The main advantage of the complexity of the pipeline is that we can reliably generate high-quality problems where we trust the answer even if the generating model can't solve the problem. For example, there are many standards (e.g. 4.MD.A.2-fraction and 4.OA.A.3) where GPT-4o scores a 0.57 and 0.80, respectively. Among the 49 skills we examined, 4o scored below a 0.90 on10 of them. While this is still relatively impressive performance, not using the symbolic grounding we do would leave us with no reliable way of calculating the solution, which we currently do programmatically using the symbolic structure.
>
> > The figures should be made in vector graphics. Right now, many plots are pixelated when zoomed in. Also, I don't recommend the use of bold font in plot as they are too distractive on the normal flow of reading.
>
> Thank you for paying such close attention to ensure the quality of our paper! We will edit all images in the final manuscript to be vector graphics, and removing bold text in the images.

---

> > ### Comment · Reviewer_Mhg7 · 2025-06-09
> >
> > Thank you for the response. I am generally satisfied with it and will maintain my score to recommend its acceptance.

---

### Official Review · Reviewer_Y1Xb · 2025-05-17

**Rating:** 6
**Confidence:** 4
**Ethics Flag:** 1

**Summary:**

The paper tries to explain why LLMs can reason with mathematical problems. It creates a dataset MathCAMPS for that purpose. There is no concrete summary, though.

**Questions To Authors:**

W1-W5

Q6. There are some grammatical and spelling errors (L:248, etc.).

Q7. The data is generated by GPT4-o. Chances are most of the generation follows a similar pattern. Is it covering enough stylistic variations in the problems generated?

**Reasons To Accept:**

S1. A new and extensive dataset for Mathematical Reasoning to do away with any contamination of reasoning datasets.

S2. Detailed analysis showing how the LLMs perform on reasoning tasks at zero-shot and after instruction-tuning.

S3. The idea of follow up questions to understand the reasoning ability of LLMs.

S4. The paper is well-written.

**Reasons To Reject:**

W1. For any benchmark dataset human evaluation is necessary to check the quality of the dataset. Even though there is automatic checking of the dataset but it would have been helpful if there is also a human evaluation.

W2. The dataset contains numbers whose addition will lead 20. This could have been extended for the curation of a richer dataset.

W3. I would have really liked to see the performance of LLMs according to digits. Do they perform better where addition results in a single digit number like (1+5) compared to two digits (8+9).

W4. How is the effect of few-shot prompting in these cases? Would that have been helpful?

W5. A short survey on how humans actually solve this problem would also have made the paper stronger specifically for the follow-up questions.

---

> ### Author Response · Authors · 2025-05-31
>
> We thank the reviewer for their comments on our paper!
>
> > W1. For any benchmark dataset human evaluation is necessary to check the quality of the dataset. Even though there is automatic checking of the dataset but it would have been helpful if there is also a human evaluation.
>
> We agree that a human evaluation is necessary, and we have indeed conducted a manual evaluation of 245 problems from MathCAMPS, as discussed in Appendix F. We see that after cycle-consistency, 97.7% of the problems in our dataset are reliable (i.e. are unambiguous and are associated with a correct answer). Thus, our manual evaluation found the MathCAMPS generation pipeline to be highly reliable.
>
> > W2. The dataset contains numbers whose addition will lead 20. This could have been extended for the curation of a richer dataset.
>
> The dataset covers 44 different skills across grades K-8, as described in our Abstract, Related Works, and Methods (Section 3.1, specifically). We will also edit the contributions to add this. Figure 1 shows the CC standard you mention simply as just a single example, but the dataset is much more comprehensive than that particular skill – the list of all Common Core standards we have implemented is given in Tables 1 through 9 in the Appendix (starting in page 18 of the submission).
>
> > W3. I would have really liked to see the performance of LLMs according to digits. Do they perform better where addition results in a single digit number like (1+5) compared to two digits (8+9).
>
> We appreciate the reviewer’s suggestion, and have done an experiment to examine this. To explore the relationship between performance and the size of the numbers involved in the problem, we plotted the accuracy for each training checkpoint for the different numbers of digits seen in the answers.
>
> Plot for accuracy vs. number of digits: https://ibb.co/TBdVtFhr
>
> We will be conducting a deeper analysis in the final manuscript, but interestingly, we note how model behavior doesn’t follow exactly what is expected. For example, even though problems with 1-digit answers are easier on average than those with two digit answers, Amber seems to consistently over perform in cases where there are 2-digit answers. This holds true for the final Pythia model as well, but not for most other training checkpoints we looked at. We thank the reviewer for their suggestion again, and are looking forward to probing this deeper to include in the final manuscript.
>
> > W4. How is the effect of few-shot prompting in these cases? Would that have been helpful?
>
> As discussed at the beginning of Section 3.2 (Lines 177-178), we employed few-shot prompting to demonstrate model behavior. We found this to be important, especially since we ran experiments on pre-training checkpoints that aren’t instruction-tuned. These were the two shots we used:
>
>
> ```
>     {"role": "system", "content": "You are a mathematics teacher that solves all problems correctly and explains your reasoning. Write your final answer in the last line of your response."},
>
>     {"role": "user", "content": "Natalia sold clips to 48 of her friends in April, and then she sold half as many clips in May. How many clips did Natalia sell altogether in April and May?"},
>
>     {"role": "assistant", "content": "Reasoning: Natalia sold 48/2 = <<48/2=24>>24 clips in May.\nNatalia sold 48+24 = <<48+24=72>>72 clips altogether in April and May.\nAnswer: 72"},
>
>     {"role": "user", "content": "Weng earns $12 an hour for babysitting. Yesterday, she just did 50 minutes of babysitting. How much did she earn?"},
>
>     {"role": "assistant", "content": "Reasoning: Weng earns 12/60 = <<12/60=0.2>>0.2 per minute.\nWorking 50 minutes, she earned 0.2 x 50 = <<0.2*50=10>>10.\nAnswer: 10"}
> ```
>
> > W5. A short survey on how humans actually solve this problem would also have made the paper stronger specifically for the follow-up questions.
>
> We agree that human experiments would be certainly interesting, especially for potential educational applications of our work. However, we believe this is a separate endeavor that is worth doing right — we would need to recruit a controlled population of humans of appropriate grade levels in order to draw informative insights (since, for instance, grade 5 students will certainly differ significantly from MTurk/Prolific workers which would be easier to recruit but perhaps less interesting for this purpose). While these evaluations would go beyond the scope of the current work, we will expand on our discussion into what these would entail for future work in our conclusion.
>
> > Q6. There are some grammatical and spelling errors (L:248, etc.).
>
> We really appreciate you pointing this out! We will make sure to edit this error and others we've caught in the final manuscript.

---

> > ### Comment · Reviewer_Y1Xb · 2025-06-01
> > **Acknowledgment**
> >
> > I acknowledge the authors' responses.
> >
> > However, the website that they pointed to is not loading.
> > Can the main results be pasted right here?

---

> > > ### Comment · Reviewer_Y1Xb · 2025-06-01
> > > **Few-shot?**
> > >
> > > I do not understand the few-shot prompts, though.
> > >
> > > What I intended was that the prompt will contain a few examples of a complete working of a problem.
> > >
> > > Are the role-assistant ones the few-shot prompts? What are the other prompts signifying?

---

> > ### Author Response · Authors · 2025-06-04
> > **Results for Digit Accuracy**
> >
> > We are really sorry about the previous link not working. The results are a graph image showing accuracy across different numbers of digits over training, which OpenReview doesn't allow us to directly upload. However, they should be viewable in the anonymized repository here: https://anonymous.4open.science/r/colm-D8D6/answer_digit_accuracy.png
> >
> > We notice that model behavior does not always follow what is expected. For example, it is expected that problems with 1-digit answers are easier on average than those with 2-digit answers. Surprisingly, we see that starting at 15% of the way through training, Amber starts to perform better on problems with 2-digit answers as compared to those with1-digit answers. We also see this in Pythia in the last decile of training, although the effect is lesser pronounced. MathCAMPS enables many analyses like this that specify where LLMs behave unconventionally, helping researchers create better hypotheses for training recipes.

---

> > > ### Comment · Reviewer_Y1Xb · 2025-06-08
> > > **Images**
> > >
> > > Thank you. I can see the images now.

---

> > ### Author Response · Authors · 2025-06-04
> > **Few-shot prompts**
> >
> > Sorry about any confusion! We employ few-shot prompting at two stages. First, we employ 2-shot prompting during problem generation. For each Common Core standard, we manually created 2 examples of how a symbolic structure specific to that standard would be converted to a word problem. This was given to GPT to demonstrate how natural language problems are written from symbolic structures.
> >
> > The second point we used few-shot prompting was while evaluating the model. The JSON-style shown above was just used as a unifying style to locally store the prompts. When we actually prompted the model, the prompt would have more like this, where it included a few examples of a complete working of both problems:
> >
> > ```
> > You are a mathematics teacher that solves all problems correctly and explains your reasoning. Write your final answer in the last line of your response.
> >
> > Natalia sold clips to 48 of her friends in April, and then she sold half as many clips in May. How many clips did Natalia sell altogether in April and May?
> >
> > Reasoning: Natalia sold 48/2 = <<48/2=24>>24 clips in May.\nNatalia sold 48+24 = <<48+24=72>>72 clips altogether in April and May. Answer: 72
> >
> > Weng earns $12 an hour for babysitting. Yesterday, she just did 50 minutes of babysitting. How much did she earn?
> >
> > Reasoning: Weng earns 12/60 = <<12/60=0.2>>0.2 per minute.\nWorking 50 minutes, she earned 0.2 x 50 = <<0.2*50=10>>10. Answer: 10
> > ```
> > Then, we appended a question from MathCAMPS to this prompt.
> >
> > Sorry about the lack of clarity about prompting! Please let us know if you have additional questions, and thank you for engaging in discussion!

---

> > > ### Comment · Reviewer_Y1Xb · 2025-06-08
> > > **Thank you**
> > >
> > > Thank you.
> > > I now understand.

---

> > ### Author Response · Authors · 2025-06-09
> >
> > We thank the reviewer for engaging with our responses, and we're glad the new results were visible and the clarifications were sufficient! Since the discussion period is ending soon (tomorrow), we'd like to know whether the reviewer has any outstanding concerns. We'll do our best to address them in time. Otherwise, if there are no remaining concerns, we'd kindly ask the reviewer to reflect this fact on their score. Thanks again for engaging with our responses!

---

> > ### Comment · Reviewer_Y1Xb · 2025-06-09
> > **Manual evaluation**
> >
> > Is it possible to give more details of the manual evaluation? What kind of problems? Who were the evaluators? Was a problem evaluated by a single person or multiple, etc.?
> >
> > Also, can these humans be asked to report how they actually solved the problem?
> >
> > I thank the authors for clarifying the other points.

---

> > > ### Author Response · Authors · 2025-06-10
> > > **Additional Information About Evaluation**
> > >
> > > Of course, thank you for your question! The manual evaluation was done on a 5% scale dataset. We generated 245 problems using our framework (5 problems/skill), and instead of discarding the non-cycle consistent ones, we just noted that they were non-cycle-consistent. Two people evaluated all the problems. The process was conducted as follows: each of the evaluators solved all 245 problems and recorded their answers. While solving the problems, the evaluators did not have insight into the final answer of the problem and whether it passed cycle consistency. The evaluator-calculated answers were compared to the "ground truth" answer of the word problem obtained from the symbolic structure. Note that a problem is faithful if an evaluator's answer == the "ground truth" answer from our pipeline. From here, we grouped problems into four categories. We include the number of problems per category below:
> > >
> > > 1. Faithful and cycle-consistent problems: 208 problems. These are problems where our evaluator's and dataset's final answers agreed. These problems also passed the cycle-consistency check (i.e. high-quality problems that would be included in MathCAMPS).
> > > 2. Unfaithful but cycle-consistent problems: 5 problems. These problems passed our cycle-consistency check, but are still bad problems (because of ambiguity issues discussed further in Appendix F.1). These represent low-quality problems that were not filtered out by our pipeline, but should have been.
> > > 3. Unfaithful and non-cycle consistent problems: 25 problems. These are bad problems, and our pipeline validly identifies them as so, discarding them (Appendix F.2).
> > > 4. Faithful but non-cycle consistent problems: 7 problems. These were problems that were good quality, but were discarded because of some error in the back-translation process. These are discussed in Appendix F.3.
> > >
> > > How these problems were solved: we allowed calculator use, but as mentioned above, the evaluators did not have access to anything besides the problem statement. Let us know if you have any additional questions!

---

> > > > ### Comment · Reviewer_Y1Xb · 2025-06-10
> > > > **Thank you**
> > > >
> > > > Thank you.
> > > >
> > > > I have increased my score.

---

> ### Author Response · Authors · 2025-05-31
>
> > Q7. The data is generated by GPT4-o. Chances are most of the generation follows a similar pattern. Is it covering enough stylistic variations in the problems generated?
>
> Since the symbolic structures are generated from 44 different skills, and since we explicitly sample random themes to condition the word problem generation, we see a large variety of the types of problems in our dataset. Here are some examples of skills, associated symbolic structures that are programmatically generated, and the natural language problems they are translated into.
> Skill: 2.NBT.B.7 (adding and subtracting within 1000)
> Symbolic structure:
>
> ```
> [[var m = 278]]
> [[var f = (m + 305)]]
> [[question l = ['f']]]
> ```
>
> *Problem:* In one month, a fireplace company sold 278 traditional wood-burning fireplaces. In the same month, they also sold 305 more gas fireplaces than traditional ones. How many gas fireplaces did the company sell?
>
> ---
>
> *Skill:* 3.MD.C.8-polygon (perimeters of polygons)
>
> *Symbolic structure:*
> ```
> [[var 274 = (((((54 + 51) + v) + 5) + 22) + 93)]]
> [[question c = ['v']]]
> ```
>
> *Problem:* A polygon has sides measuring 54cm, 51cm, 5cm, 22cm, and 93cm. If the total perimeter of the polygon is 274cm, what is the length of the sixth side?
>
> ---
>
> *Skill:* 8.EE.C.8 (systems of equations with 2 variables)
> *Symbolic structure:*
> ```
> [[var 37 = ((40 * s) - (91 * q))]]
> [[var 121 = ((30 * s) + (12 * q))]]
> [[question y = ['q', 's']]]
> ```
> *Problem:* Let's find the values of variables q and s by solving the following system of equations:\n\n1. (40 * s) - (91 * q) = 37\n2. (30 * s) + (12 * q) = 121.

---

> > ### Comment · Reviewer_Y1Xb · 2025-06-01
> > **Acknowledgment**
> >
> > I acknowledge the comment, and I understand the variety of problems now.

---

### Official Review · Reviewer_XbsE · 2025-05-27

**Rating:** 6
**Confidence:** 3
**Ethics Flag:** 1

**Summary:**

This paper investigates the learning dynamics of language model pre-training and instruction-tuning, with a particular focus on math reasoning. To avoid data contamination, this work introduces MathCAMPS, a synthetic benchmark of 4 900 grade-school math problems drawn from 44 Common Core standards. Problems are generated symbolically and verbalised with GPT-4. The authors then use MathCAMPS to analyse how three open-weight model families with public checkpoints (Amber, OLMo-2 7 B, Pythia-12 B) acquire mathematical skills during pre-training and how instruction-tuning perturbs those skills.

**Questions To Authors:**

- The paper infers that OLMo’s accuracy drop after tuning is due to high-quality math data in its second pre-training stage.
Can the authors provide controlled experiments that vary only the instruction-tuning corpus or objective to isolate which factor causes degradation?

- Some CC standards involve prerequisite relationships (e.g., fraction addition before rational-number operations). Did the authors examine whether mastering one skill accelerates learning of a downstream skill, and could this be used to design better curricula for model training?

**Reasons To Accept:**

- Fine-grained, curriculum-grounded benchmark. By tying each item to an explicit Common Core skill, MathCAMPS enables analyses that GSM8K/MATH cannot—e.g., plotting learning curves per skill and grade cluster.

- Fully automated generation with quality safeguards. The grammar-based sampling plus cycle-consistency filter removes ~83 % of unfaithful problems, yielding 97.7 % faithful items in a manual audit.

- Transparent look at training dynamics. Using open checkpoints, the paper provides empirical evidence that skill acquisition is gradual and curriculum-aligned, offering data points for theorists of LLM learning.

**Reasons To Reject:**

- Incremental with respect to existing synthetic math resources. Recent work (e.g., GSMore [1], GSM-Symbolic [2], DyVal [3]) already synthesises or perturbs math problems for robustness studies; MathCAMPS differs mainly in being tied to Common Core standards, which feels like a change of packaging rather than a conceptual advance.

- Heavy reliance on GPT-4 for both generation and back-translation. Although cycle consistency helps, GPT-4 can still hallucinate or encode hidden shortcuts, and no contamination check against GPT-4’s training data is provided.

- Findings are unsurprising. Earlier-grade skills appearing earlier, decimals being easier than fractions, and instruction-tuning sometimes hurting accuracy are all intuitive and have been anecdotally reported before; the paper lacks a compelling theoretical insight.

[1] Hong et al. Evaluating LLMs' Mathematical and Coding Competency through Ontology-guided Interventions.
[2] Mirzadeh et al. GSM-Symbolic: Understanding the Limitations of Mathematical Reasoning in Large Language Models.
[3] Zhu et al. DyVal: Dynamic Evaluation of Large Language Models for Reasoning Tasks.

---

> ### Author Response · Authors · 2025-05-31
>
> We thank the reviewer for their detailed comments and suggestions! We have responded to the questions/added some clarifications below.
>
> > Incremental with respect to existing synthetic math resources. Recent work (e.g., GSMore [1], GSM-Symbolic [2], DyVal [3]) already synthesises or perturbs math problems for robustness studies; MathCAMPS differs mainly in being tied to Common Core standards, which feels like a change of packaging rather than a conceptual advance.
>
> We agree that our main difference, in terms of the dataset, is exactly its close tie to a human curriculum. However, that alone enables a whole suite of analyses (which we've explored in the paper) that would be impossible to conduct on any of the existing datasets. Moreover, the follow-up question generation we introduced (Section 3.3, used to answer RQ3) is not present in any of the existing mathematical reasoning datasets we know of, synthetic or otherwise, including GSMore, GSM-Symbolic and DyVal, GSM8k, MATH and many others.
>
> > Heavy reliance on GPT-4 for both generation and back-translation. Although cycle consistency helps, GPT-4 can still hallucinate or encode hidden shortcuts, and no contamination check against GPT-4’s training data is provided.
>
> We agree that checking how much the dataset is influenced by the choice of GPT-4 as the generator model is important. Thus, to address this, we conducted an analysis where we generated a dataset using the exact same pipeline, but with Claude instead. We found that the problem difficulty seemed mostly unaffected by the model used for generation. For instance, GPT-4o achieves an accuracy of 0.910 on the GPT-4-generated dataset, and an accuracy of 0.954 on the Claude generated dataset. Meanwhile, Claude scores 0.887 on the GPT-4-generated dataset, and 0.909 on the Claude-generated dataset. Thus, in both cases, the consistent result is that GPT-4o outperformed Claude indicating that the generation model is not necessarily generating problems that are easier for itself (e.g., by copying from its training data, or introducing shortcuts). We will be adding Appendix G to the final paper to include the full analysis of this result, which will be present in the final version of the paper.
>
> > The paper infers that OLMo’s accuracy drop after tuning is due to high-quality math data in its second pre-training stage. Can the authors provide controlled experiments that vary only the instruction-tuning corpus or objective to isolate which factor causes degradation?
>
> Thank you for the thoughtful suggestion! We agree that controlled experiments isolating the effects of the instruction-tuning data corpus would be useful in isolating how various factors within the data impact the final performance of the model. While we were not able to conduct such experiments due to the high computational cost of fine-tuning LLMs, our current works take a first step by analyzing the impact of post-training on three publicly available models (Amber, Llama, and OLMo), each of which underwent a distinct post-training schema. This cross-model comparison highlights general trends and introduces hypotheses we hope future works can explore in more controlled manners.
>
> Our results are also in alignment with an observation made by the very recent work of Springer et. al. [1], which considers the quantity of data seen during the pre-training of the LLM and its instruction-tuned counterpart’s performance. They show that the more tokens an LLM sees during pre-training, the less effective instruction-tuning is on it. In our case, OLMo and Llama, which each saw 15T and 4T tokens of data during pre-training respectively, saw more degradation post fine tuning compared to Amber, which saw ~1.3T tokens during pre-training. We will add a brief discussion of Springer et. al.’s paper as a possible explanation in our final manuscript.
>
> [1] J. M. Springer et. al. Overtrained Language Models Are Harder to Fine-Tune (https://arxiv.org/abs/2503.19206)

---

> > ### Comment · Reviewer_XbsE · 2025-06-11
> >
> > Thanks for your response. Most of my concern has been solved. I will raise my score accordingly.

---

### Decision · Program_Chairs · 2025-07-08

**Decision:**

Accept

**Comment:**

This paper presents a novel investigation into how mathematical reasoning skills emerge in LLMs during both pre-training and instruction tuning. The authors introduce MathCAMPS, a synthetic, curriculum-aligned benchmark dataset grounded in 44 Common Core math skills from K–8. Using MathCAMPS and a set of open-weight LLMs, the authors examine the temporal progression of skill acquisition and the impact of instruction tuning. Notably, they find that LLMs tend to acquire skills in a curriculum-correlated order despite randomized data, and that instruction tuning can sometimes degrade certain math abilities.

Strengths:
1. The dataset is carefully constructed with symbolic reasoning, back-translation, and cycle-consistency checks.
2. The paper offers the first analysis of when and how mathematical abilities emerge during pretraining. Observing skill progression and degradation across checkpoints is valuable for both developers and theorists.
3. The comparison across models provides intriguing evidence that instruction tuning can harm performance, especially in overtrained models. This opens important questions for post-training strategies.

Weaknesses:
1. While MathCAMPS is carefully crafted, some reviewers saw it as a repackaging of ideas already present in datasets like GSM-Symbolic or DyVal. The differentiator here is its tight alignment with curriculum standards, which enables new analyses – but this may not feel groundbreaking to all.
2. The instruction tuning experiments are limited to pre/post checkpoints rather than step-wise tracking. While this is understandable due to compute constraints, some deeper ablation would have strengthened causal claims.
3. The generation pipeline heavily depends on GPT-4o, though authors attempted to mitigate this by replicating part of the pipeline with Claude.

Overall, this paper presents a meaningful and well-executed contribution to the understanding of how LLMs acquire mathematical reasoning skills. It blends dataset development, training dynamics analysis, and educational relevance in a way that is rarely done together. Despite some incremental aspects and practical limitations, the novel insights, transparency, and the quality of the work make it a strong candidate for publication.